



# Stochastic simulation of streamflow and spatial extremes: a continuous, wavelet-based approach.

Manuela I. Brunner[1] and Eric Gilleland[1]

[1]Research Applications Laboratory, National Center for Atmospheric Research, Boulder CO, USA

**Correspondence:** Manuela I. Brunner (manuelab@ucar.edu)

**Abstract.** Stochastically generated streamflow time series are used for various water management and hazard estimation applications. They provide realizations of plausible but yet unobserved streamflow time series with the same temporal and distributional characteristics as the observed data. However, the representation of non-stationarities and spatial dependence among sites remains a challenge in stochastic modeling. We investigate whether the use of frequency-domain instead of time-domain

models allows for the joint simulation of realistic, continuous streamflow time series at daily resolution and spatial extremes at multiple sites. To do so, we propose the stochastic simulation approach called Phase Randomization Simulation using wavelets *PRSim.wave* which combines an empirical spatio-temporal model based on the wavelet transform and phase randomization with the flexible four-parameter kappa distribution. The approach consists of five steps: (1) derivation of random phases, (2) fitting of kappa distribution, (3) wavelet transform, (4) inverse wavelet transform, and (5) transformation to kappa distribu-

tion. We apply and evaluate *PRSim.wave* on a large set of 671 catchments in the contiguous United States. We show that this approach allows for the generation of realistic time series at multiple sites exhibiting short- and long-range dependence, non-stationarities, and unobserved extreme events. Our evaluation results strongly suggest that the flexible, continuous simulation approach is potentially valuable for a diverse range of water management applications where the reproduction of spatial dependencies is of interest. Examples include the development of regional water management plans, the estimation of regional

flood or drought risk, or the estimation of regional hydropower potential among others.

**Keypoints:**

1. Stochastic simulation of continuous streamflow time series using an empirical, wavelet-based, spatio-temporal model in combination with the parametric kappa distribution.

2. Generation of stochastic time series at multiple sites showing temporal short- and long range dependence, non-stationarities,
and spatial dependence in extreme events.

3. Implementation of *PRSim.wave* in R-package *PRSim*: Stochastic Simulation of Streamflow Time Series using Phase Randomization.





## 1   Introduction

Stochastic models are used to generate long time series or large event sets showcasing the full variability of a phenomenon. In
hydrology, we use stochastically generated time series or event sets to refine water management plans, to get a better idea of
potential reservoir inflows, or to develop suitable adaptation strategies for droughts and floods. If the focus is on such extreme
events, event-based instead of continuous simulation approaches are often employed (e.g. Bracken et al., 2016; Diederen et al.,
2019; Quinn et al., 2019). This strategy requires an *a priori* definition of extreme events and leads to a loss of temporal
information, e.g. on the season of occurrence. In contrast, continuous simulation approaches allow for the simulation of time
series including, but not limited to, extreme events which are provided together with their time of occurrence. Such continuous
approaches enable the investigation of drought and flood characteristics going beyond magnitude and extending to timing,
duration, or volume. However, the model representation of this additional information on timing adds some challenges because
the temporal characteristics of the data need to be represented in addition to their distribution. These temporal characteristics
include fluctuations on short and long timescales (Rajagopalan et al., 2010) and potential non-stationarities in the data (Nowak
et al., 2011).

There exists a variety of continuous, stochastic modeling approaches which differ in their capability of representing dis-
tributional and/or temporal characteristics in the data. Here, we focus on direct modeling approaches that directly simulate
streamflow using a stochastic model as opposed to indirect approaches which use a hydrological model to transform stochas-
tically generated precipitation to streamflow. The most commonly used stochastic simulation approaches belong to the two
classes of parametric and nonparametric models. Parametric models include autoregressive moving average (ARMA) models
and their modifications (Stedinger and Taylor, 1982; Papalexiou, 2018), and temporal disaggregation models such as fractional
Gaussian noise models (Mandelbrot, 1965), fast fractional Gaussian noise models (Mandelbrot, 1971), broken line models
(Mejia et al., 1972), and fractional autoregressive integrated moving average models (Hosking, 1984). Nonparametric mod-
els are based on disaggregation and resample from the data with perturbations and include kernel density estimation (Lall
and Sharma, 1996; Sharma et al., 1997) and various bootstrap approaches such as simple bootstrap, moving block-bootstrap,
nearest-neighbor bootstrap (Salas and Lee, 2010; Herman et al., 2016), matched-block bootstrap (Srinivas and Srinivasan,
2006), or maximum-entropy bootstrap (Srivastav and Simonovic, 2014). These parametric and non-parametric methods have
different strengths and weaknesses as e.g. discussed in Rajagopalan et al. (2010) but none of these time domain methods can
capture the spectral properties of the observed time series (Erkyihun et al., 2017). Furthermore, these time-domain models
struggle with the representation of spatial dependence. For parametric models, the number of parameters grows rapidly with
the number of locations (Caraway et al., 2014) and the spatial dependence structure is similar for high, medium, and low flow
values, which is not the case for observed data (Sharma et al., 1997). Similarly, non-parametric approaches are not effective
for multiple-site streamflow generation because of the high dimension of the disaggregation problem (Nowak et al., 2010).

In contrast to time-domain models, frequency-domain models allow for the simulation of surrogate data with the same
Fourier spectra as the raw data (Theiler et al., 1992) and can easily be extended to multiple sites (Prichard and Theiler, 1994;
Schreiber and Schmitz, 2000). Such methods are based on the randomization of the phases of the Fourier transform and are



known as the amplitude-adjusted Fourier transform (AAFT) (Lancaster et al., 2018). They have only recently been used in hydrology for other applications besides hypothesis testing, trend detection (Radziejewski et al., 2000), and the identification of nonlinearities in time series (Schmitz and Schreiber, 1996; Kugiumtzis, 1999; Venema et al., 2006; Maiwald et al., 2008).

Serinaldi and Lombardo (2017) used an iterative AAFT method to generate binary series of rainfall occurrence and non-occurrence. Brunner et al. (2019) used a phase randomization approach in combination with the flexible four-parameter kappa distribution (Hosking, 1994) to simulate continuous discharge time series including unobserved extremes. The approach is implemented in the R-package Stochastic Simulation of Streamflow Time Series using Phase Randomization *PRSim* (Brunner and Furrer, 2019) and can be applied to both individual and multiple sites. This phase randomization approach has been shown

to reproduce the distributional and temporal characteristics of the data at individual sites well (Brunner et al., 2019; Brunner and Tallaksen, 2019). However, the approach has some deficiencies when applied to multiple sites because spatial dependencies in both daily discharge and extreme events are underestimated. In addition, the approach does not allow for the consideration of non-stationarities.

In contrast to the Fourier transform, the wavelet transform allows for the representation of non-stationarities in time series

(Rajagopalan et al., 2010). For a short introduction to the wavelet transform see Sect. 2. In addition, it may help to improve the representation of spatial dependencies because it does not require a transformation to the normal distribution and back to the original, skewed distribution, which usually weakens spatial correlations (Embrechts et al., 2010). Because of its favorable properties, the wavelet transform has been used in stochastic time series generation in various ways. Kwon et al. (2007) proposed a wavelet-based autoregressive modeling approach (WARM) suitable for systems with a quasi-periodic long

memory behavior. They used the continuous wavelet transform to decompose a time series into several statistically significant components. Each of these components was fitted using a linear AR model which was subsequently used for simulation. Later, Nowak et al. (2011) adapted this WARM approach such that it can handle non-stationarities. Another possibility of handling non-stationarities is the wavelet-based time series bootstrap model introduced by Erkyihun et al. (2016) which generates wavelet-derived signal components with a block resampling approach therefore replacing the AR component of the WARM.

An alternative to these approaches where only certain signal components are modified are approaches that randomize the wavelet coefficients for all components. These approaches typically perform a discrete wavelet decomposition, randomize the wavelet coefficients (i.e. amplitudes) and then invert the transform to produce a new realisation of a time series (Breakspear et al., 2003; Keylock, 2007; Wang et al., 2010; Lancaster et al., 2018). However, a completely random shuffling of coefficients destroys their periodicity. To overcome this drawback, Breakspear et al. (2003) used block resampling on coefficients and

Keylock (2007) introduced a threshold criterion that is used to pin particular wavelet coefficients to the same position in the wavelet domain for both the original and the surrogate data. Despite these workarounds, some of the long-term periodicities and/or non-stationarities may not be preserved (Breakspear et al., 2003). Using a continuous instead of a discrete wavelet transform can prevent this loss. To generate non-stationary surrogate time series, Chavez and Cazelles (2019) extended the classic phase-randomised surrogate data algorithm to the time-frequency domain using a dataset of weekly measles notifications and

an electroencephalographic recording. They randomized the phases of the wavelet transform instead of the real valued amplitudes. This approach has the advantage of being non-parametric, avoids assumptions on the distribution and dependence





structure of the data, allows for multiple realizations, and can be extended to multiple sites by randomizing the phases for multiple time series in the same way.

We investigate whether such a wavelet-based phase randomization approach allows for a realistic representation of spatial dependence in both continuous streamflow time series and spatial extremes. To do so, we propose a continuous wavelet-based approach for the stochastic generation of streamflow time series, hereafter referred to as *PRSim.wave* which is based on the empirical spatio-temporal model used by Chavez and Cazelles (2019), i.e. wavelet-based phase randomization. This empirical approach is supposed to overcome difficulties in modeling spatial dependence over a large domain as occurs when using parametric models (Caraway et al., 2014). We combine this empirical spatio-temporal model with a parametric distribution to enable the generation of hydrological extremes exceeding the range of the observed values. By doing so, we extend the approach by Brunner et al. (2019) from the Fourier to the wavelet domain therefore improving the representation of spatial dependencies and non-stationarities.

We implement *PRSim.wave* in the R-package *PRSim* (Brunner et al., 2019) and apply and evaluate it on a large dataset of 671 catchments in the contiguous United States. Our evaluation results indicate that the flexible, continuous simulation approach can be used for a diverse range of water management applications requiring continuous discharge time series at multiple sites or for regional drought and flood hazard assessments not limited to peak magnitude or maximum deficit but extending to event duration and volume.

## 2 Theoretical background

Wavelet decomposition transforms a one-dimensional time series to a two-dimensional time-frequency space (Torrence and Compo, 1998) using either a discrete or continuous wavelet transform. Both transforms decompose a hydrological series into a set of coefficients, representing different scales or frequency bands (Sang, 2012). The coefficients of the discrete transform are real numbers representing amplitudes. In contrast, the coefficients derived from the continuous transform have a real and an imaginary part corresponding to an amplitude and phase, respectively. This additional information on the phases makes complex wavelet functions more suitable for capturing oscillatory behavior (Torrence and Compo, 1998).

The wavelet function used for the transform should reflect the features present in the time series. Because of its smooth features, the Morlet wavelet has often been used in hydrological applications (Schaefli et al., 2007) and is given by (Torrence and Compo, 1998) :

$$\psi_0(\eta) = \pi^{-1/4} e^{i\omega_0 \eta} e^{-\eta^2/2}. \tag{1}$$

The continuous wavelet transform is defined as the convolution of a time series $x_n$ of length $n$ with a scaled version of $\psi_0(\eta)$:

$$W_n(h) = \sum_{n'=0}^{N-1} x_{n'} \psi^* \left[ \frac{(n'-n)\delta t}{h} \right], \tag{2}$$

where the (*) indicates the complex conjugate. Varying the wavelet scale $h$ and translating along the localized time index $n$ allows for showing the amplitude of certain features versus scale and how the amplitude varies with time and scale. An inverse





filter can be used to reconstruct the original time series as the sum of the real part of the wavelet transform over all scales $(h_1, ..., h_J)$:

$$125 \quad x_n = \frac{\delta j \delta t^{1/2}}{C_\delta \psi_0(0)} \sum_{j=0}^{J} \frac{R\left(W_n(h_j)\right)}{h_j^{1/2}}, \tag{3}$$

where the factor $\psi_0(0)$ removes the energy scaling, $h_j^{1/2}$ converts the wavelet transform to an energy density, and the factor $C_\delta$ is a constant for each wavelet function (0.776 for the Morlet wavelet).

## 3 Data and Methods

We develop and apply the stochastic simulation approach *PRSim.wave* using a large-scale dataset of 671 stations in the contiguous United States (CONUS). We evaluate the approach with respect to distributional and temporal characteristics at individual sites and with respect to spatial dependencies across multiple sites in general and for floods in particular.

### 3.1 Study area

The 671 catchments in the United States cover a wide range of discharge regimes minimally influenced by human activity (Newman et al., 2015). Daily discharge data were downloaded for the period 1981–2018 from the USGS water information system (USGS, 2019) using the R-package dataRetrieval (De Cicco et al., 2018).

For illustration and validation purposes, we select three regions which are distinct in terms of their hydrological regimes and their flood behavior: (1) catchments in the lower-elevation coastal Pacific Northwest characterized by high mean annual precipitation and a strong discharge seasonality experiencing floods mainly in December and January (2) catchments in Texas with low mean discharge, weak seasonality, and flood occurrence in spring to fall, and (3) catchments in the Mid-Atlantic coastal plain and central Appalachian Mountains with a strong streamflow seasonality showing flood occurrence for much of the year except early-mid summer. For each of these regions, four catchments are chosen for illustration purposes (Fig. 1).

### 3.2 Simulation procedure

The stochastic simulation procedure *PRSim.wave* for multiple sites consists of 5 main steps (Fig. 2), which can be run $p$ times to generate $p$ spatially consistent time series over $n$ sites at daily resolution:

1. **Derivation of random phases**: A random discharge time series (white noise) of the same length as the input series is sampled from a normal distribution with mean 0 and standard deviation 1. The wavelet transform is applied to the white noise series using the Morlet wavelet (Eq. (1)) for $h = 100$ wavelet scales. The phases of the wavelet transform are derived. These same phases are used for all the sites considered to retain spatial dependencies among sites.

2. **Fitting of kappa distribution**: The flexible four-parameter kappa distribution (Hosking, 1994) is fit to the daily values of the observed input time series using L-moments. These daily distributions will be used for the transformation in Step





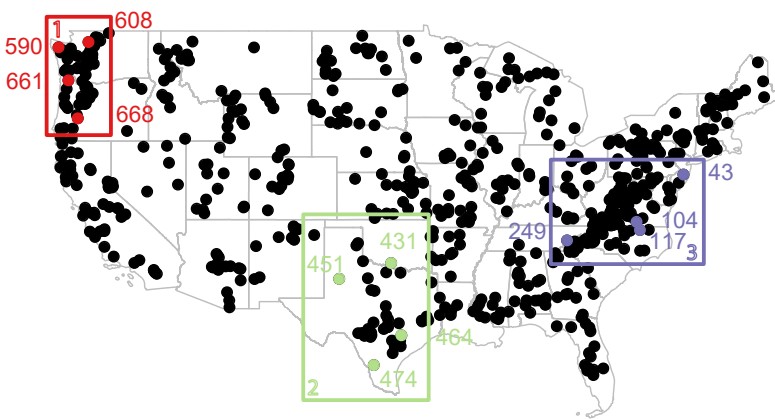

**Figure 1.** Location of 671 stations in the CAMELS dataset and of four catchments chosen per example region: (1) Pacific Northwest (red; 590, 608, 661, and 668), (2) Texas (light green; 431, 451, 464, and 474), and (3) Mid-Atlantic (purple; 43, 104, 117, 249).

5 to simulates extreme values going beyond the empirical distribution. The cumulative distribution function of the kappa distribution is expressed as

$$F(x) = \left\{ 1 - \xi[1 - \kappa(x - \mu)/\sigma]^{1/\kappa} \right\}^{1/\xi}, \tag{4}$$

where $\mu$ is the location parameter, $\sigma$ is the scale parameter which must be positive, and $\kappa$ and $\xi$ are the shape parameters

(R-package homtest Viglione, 2009).

The kappa distribution was found suitable for fitting observed streamflow data in U.S. catchments (Blum et al., 2017). A suitable fit is also found for our data as confirmed by the Kolmogorov–Smirnov and Anderson–Darling tests (Chernobai et al., 2015) which did not reject the null hypothesis at $\alpha = 0.05$ for most catchments. We fit a separate distribution for each day to take into account seasonal differences in the distribution of daily streamflow values. To do so, we use the

daily values in a 30-day window around the day of interest. This procedure guarantees a large enough sample for the parameter fitting procedure, and allows for smoothly changing distributions along the year. For leap years, flows from February 29 are removed to maintain constant sample sizes across years as in Blum et al. (2017). In a few regions with many zero discharge values (e.g. some catchments in the Great Plains) fitting the kappa distribution is not possible and we therefore use the empirical distribution instead. After fitting the daily distributions, the mean discharge is subtracted

from the observed discharge values to center the observations.





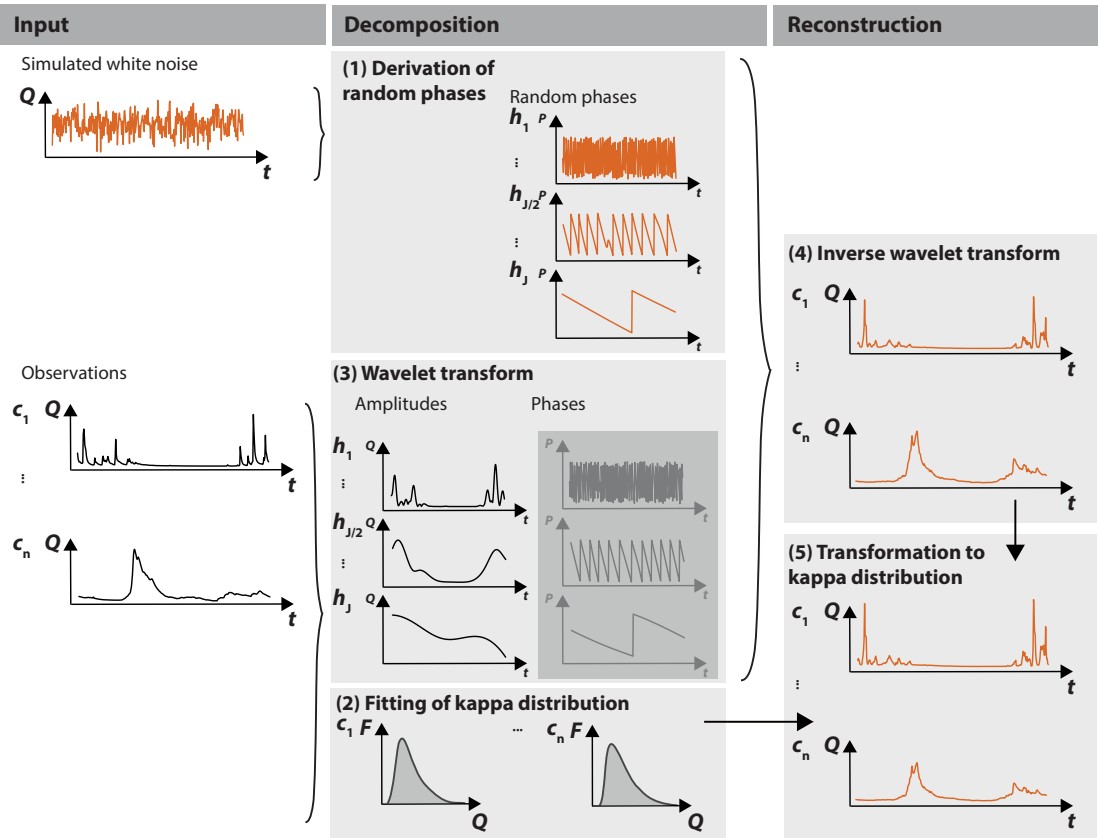

**Figure 2.** Illustration of stochastic simulation approach *PRSim.wave*: (1) Derivation of random phases using a white noise time series to be used for the simulation at all stations; for each station: (2) Fitting of kappa distribution to the observed streamflow time series; (3) Wavelet transform to derive the amplitudes and phases of the time series; (4) Inverse wavelet transform recomposing a time series using the random phases from Step 1 in combination with the amplitudes of Step 3; and (5) Transformation to the kappa distribution using the kappa distribution fitted in Step 2. Steps 1–5 are repeated $p$ times to generate $p$ time series.

3. **Wavelet transform**: The wavelet transform (Eq. (2)) is applied to the input time series using the complex-valued Morlet wavelet (Eq. (1)) to derive the amplitudes for $h = 100$ wavelet scales.

4. **Inverse wavelet transform**: Reconstruction of the series in the time domain by using the inverse wavelet transform (Eq. (3)) combining the random phases derived in Step 1 and the amplitudes derived in the previous step.

5. **Transformation to kappa distribution**: The simulated values are transformed to the kappa domain using the fitted daily kappa distributions from Step 2. For each day, a random sample is generated from the fitted, daily kappa distribution. The simulated values are replaced by the values generated from the kappa distribution using rank-ordering. This procedure is repeated for each day in the year.





Step 1 is run only once per iteration to maintain spatial dependencies in the data while Steps 2–5 are run for each station
separately.

### 3.3 Evaluation

We run the stochastic simulation algorithm for the 671 catchments in the dataset $n = 100$ times to generate 100 time series of
the same length as the observed time series, i.e. 38 years. We look at (1) individual sites to evaluate the general distributional
and temporal correlation characteristics as well as the reproduction of high- and low flows; (2) three sets of four stations each
to evaluate the spatial consistencies in daily discharge and floods: Pacific Northwest, Texas, and Mid-Atlantic (Fig. 1); and (3)
at the set of 671 catchments to evaluate spatial consistencies in floods across large scales.

The evaluation at individual sites encompasses a comparison of observed and simulated distributional and temporal discharge
characteristics. The distributional characteristics considered are the mean annual hydrograph showing variation of flow with
season, three years of daily data illustrating the overall behavior of the series, the seasonal distributions (winter: Dec–Feb,
spring: Mar–May, summer: June–Aug, fall: Sept–Nov), and monthly mean, maximum and minimum values. The temporal
characteristics considered include the autocorrelation (acf) and partial autocorrelation functions (pacf) measuring the strength
of temporal dependence for different time lags, the power spectrum indicating how power varies with frequency and showing
high values at those frequencies that correspond to strong periodic components (Shumway and Stoffer, 2017), the normalized
average power per scale over all time steps indicating oscillations, and the scale-averaged wavelet power (Erkyihun et al., 2016)
for the three scales with the highest average power revealing non-stationarities in oscillations. To evaluate the capability of the
approach to simulate extreme values, we compare observed and simulated low- and high-flows, i.e., flows below or above a
threshold corresponding to the 0.05 and 0.95 percentiles, respectively.

The evaluation at multiple sites comprises both an assessment of how the general dependence structure in the data is re-
produced and an assessment of how the dependence in high and low extremes is captured. The assessment of the general
dependence structure encompasses a comparison of observed and simulated discharge time series for the catchments in the
three example regions, a comparison of pairwise observed and simulated cross-correlations for the example stations in the
Pacific Northwest region, and a comparison of variograms of the observed and simulated series across all stations (R-package
SpatialExtremes; Ribatet, 2019) given by (Cressie, 1993):

$$2\gamma(\mathbf{s}_1, \mathbf{s}_2) = var(Z(\mathbf{s}_1) - Z(\mathbf{s}_2)) = E[(Z(\mathbf{s}_1) - Z(\mathbf{s}_2))^2], \tag{5}$$

where $Z$ is a random variable (here, streamflow) measured at two locations $s_1$ and $s_2$. In order to be able to discern the shapes
of the variograms, they are first smoothed using splines.

For assessing how spatial dependencies in extremes are reproduced, we first compare observed and simulated times of
occurrences of flood events for the catchments in the three example regions. We then compare observed to simulated F-
madograms for flood events across all stations. The F-madogram is a measure of spatial dependence that compares the ordering
of extreme events between two time series of extreme events (Cooley et al., 2006) and is expressed as:

$$v^F(d) = \frac{1}{2} E |F[Z(\mathbf{s} + d)] - F[Z(\mathbf{s})]|, \tag{6}$$





where $Z(\mathbf{s})$ are transformed to have Fréchet margins so that $F(s) = \exp(-1/s)$, and $d$ is the distance between a pair of stations (R-package SpatialExtremes; Ribatet, 2019). We finally compute the tail dependence coefficient $\chi$ across all stations defined as (Coles, 2001):

$$\chi(u) = Pr\{Y > G^{-1}(u) \mid X > F^{-1}(u)\} = Pr\{V > u \mid U > u\}, \tag{7}$$

where $X$ and $Y$ are uniformly distributed random variables with distribution functions $F$ and $G$, and $u$ is a threshold (R-package extRemes; Gilleland and Katz, 2016, ). Tail dependence estimators depend heavily on the sample size and are subject to large uncertainty given the sample size of 38 years (Serinaldi et al., 2015).

## 4 Results

### 4.1 Single-site simulations

Both the distributional and temporal dependence characteristics of the time series at individual sites are well modeled as shown by comparing observed and stochastically simulated time series for the two stations on the Nehalem and Navidad rivers (Fig. 3 and 4). The hydrological regimes (3a), the overall behavior of the time series (3b–c), the seasonal (3d) and monthly distributions (3e–g) are well captured by the simulations. The simulations, however, are capable of simulating values going beyond the range of the observations as intended by using the kappa distribution. In addition to these distributional characteristics, the temporal correlation characteristics (4a–c), the oscillations in the data (4d), and the non-stationarities in these oscillations (4d) are well captured by the simulations as well.





**Figure 3.** Comparison of observed (black) and simulated (orange) distributional discharge characteristics for (i) the station Nehalem River near Foss, OR (USGS 14301000, id 661) in the Pacific Northwest and (ii) the station Navidad Rv near Hallettsville, TX (USGS 08164300, id 464) in Texas: (a) mean annual hydrographs, (b and c) observed and simulated time series for three years, (d) seasonal discharge distribution, (e) monthly mean discharge, (f) monthly maximum discharge, and (g) monthly minimum discharge.



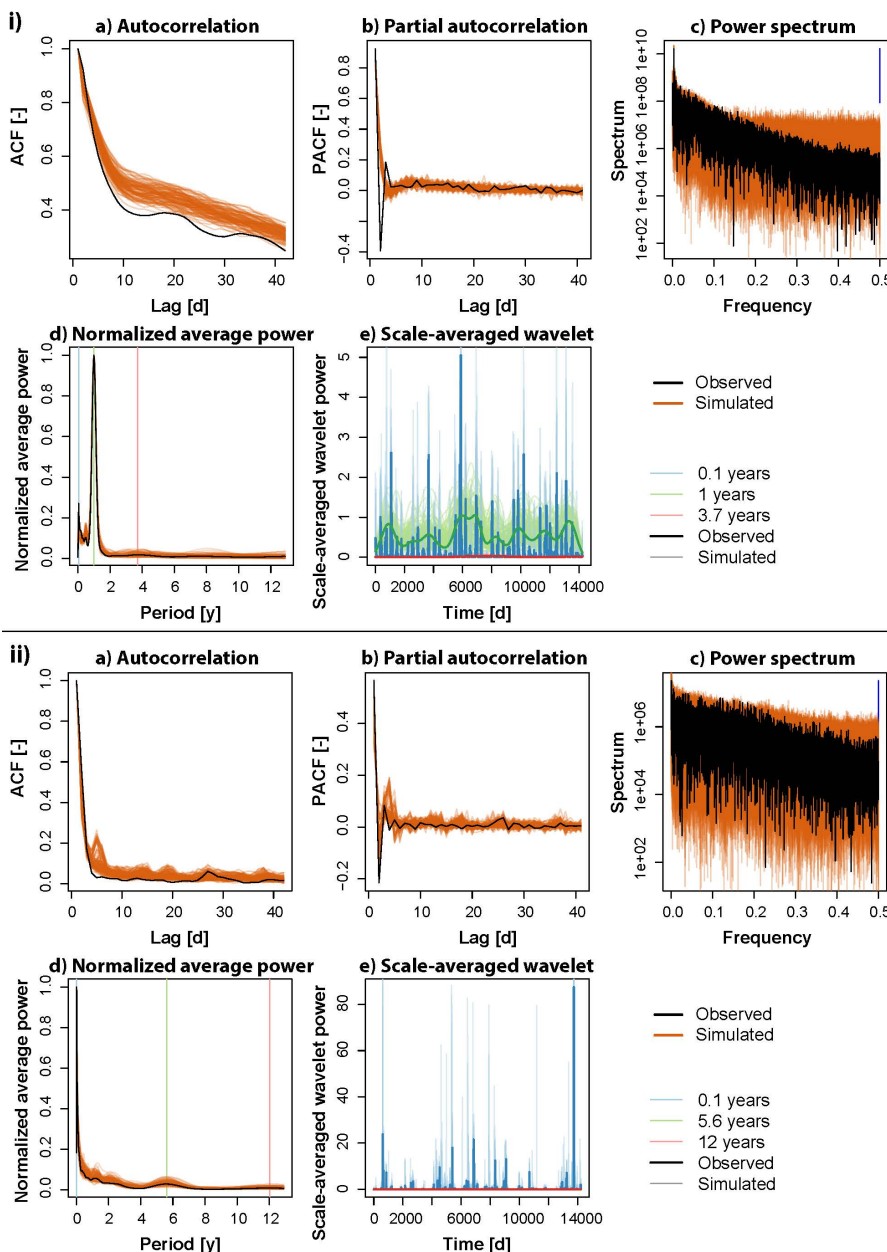

**Figure 4.** Comparison of observed (black) and simulated (orange) temporal discharge characteristics for (i) the station Nehalem River near Foss, OR (USGS 14301000, id 661) in the Pacific Northwest and (ii) the station Navidad Rv near Hallettsville, TX (USGS 08164300, id 464) in Texas: (a) acf, (b) pacf, (c) power spectrum, (d) normalized average power, and (e) scale averaged-wavelet power.





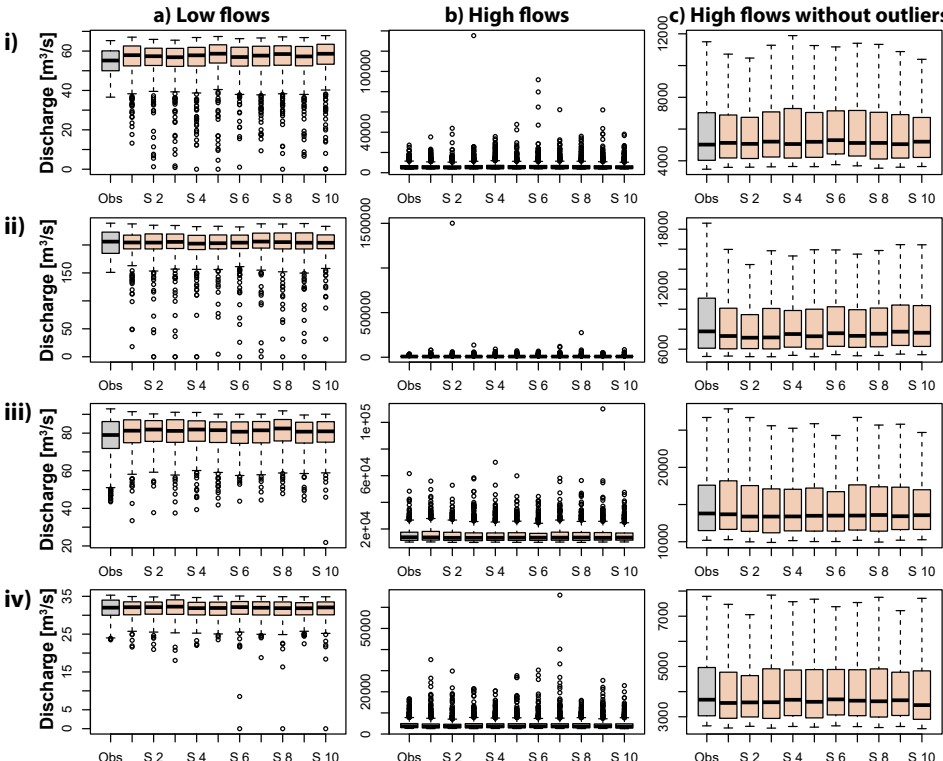

**Figure 5.** Comparison of observed (grey) (a) low- and (b–c) high-flow distributions (with and without outliers) with simulated distributions of 10 runs (orange) for the four catchments in the Pacific Northwest: (i) Calawah River near Forks, WA (USGS 12043000, id 590), (ii) Stillaguamish River near Arlington, WA (USGS 12167000, id 608), (iii) Nehalem River near Foss, OR (USGS 14301000, id 661), and (iv) Steamboat Creek near Glide, OR (USGS 14316700, id 668).

Both high- and low- extremes are realistically modeled as illustrated by the distributions of the above and below threshold events of the four catchments in the Pacific Northwest (Fig. 5). While the median of the observed low and high-flow distri-
butions is well met by the simulated medians, again the simulations allow for the generation of extreme low- and high flows going beyond the observed values because of the use of the theoretical kappa distribution.





## 4.2 Multi-site simulations

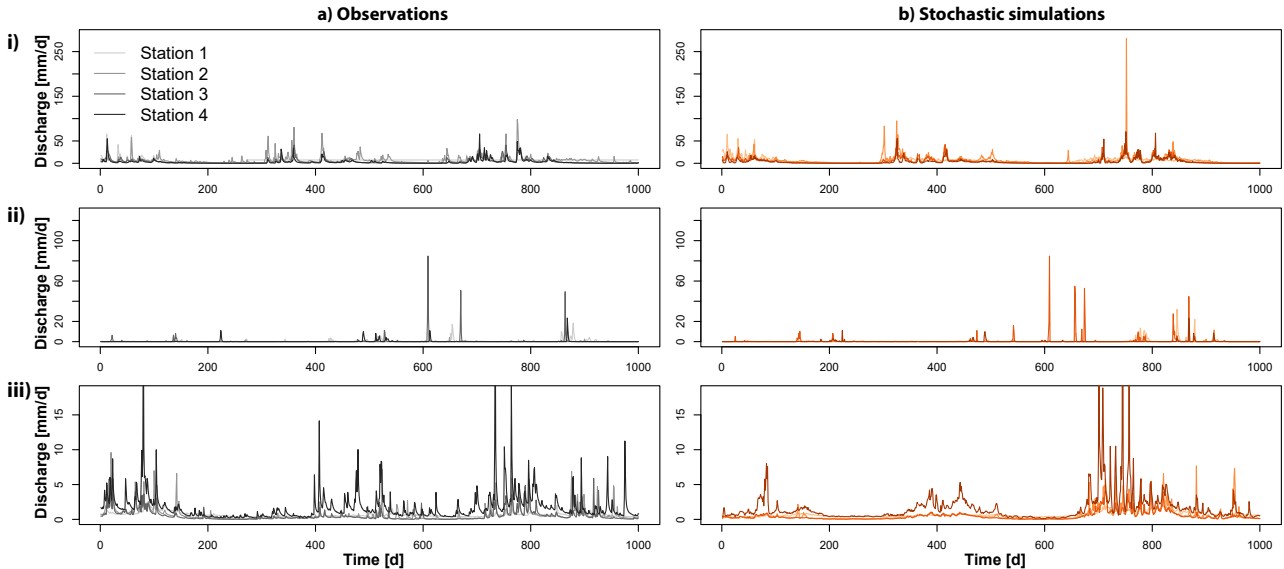

**Figure 6.** Comparison of three years of multi-site observations (a) with multi-site stochastic simulations (b) for the four catchments in the example regions (i) Pacific Northwest, (ii) Texas, and (iii) Mid-Atlantic. Each region is displayed on its own scale.

The stochastic simulation approach *PRSim.wave* not only allows for the reproduction of the distributional and temporal characteristics of time series at single sites but also for simulating spatially coherent time series at multiple sites (Fig. 6). Independent

of the region considered, the simulations realistically represent the observed behavior of the time series. This visual impression of a good performance with respect to the reproduction of spatial correlations in daily discharge data is confirmed by comparing observed and stochastically simulated cross-correlation functions for the catchments in the Pacific Northwest (Fig. 7). Both the shape and magnitude of the cross-correlation functions are well simulated. The good performance in terms of reproducing the general spatial dependence structure in the data can be generalized to other regions as shown by a comparison of observed

and simulated variograms (Fig. 8).



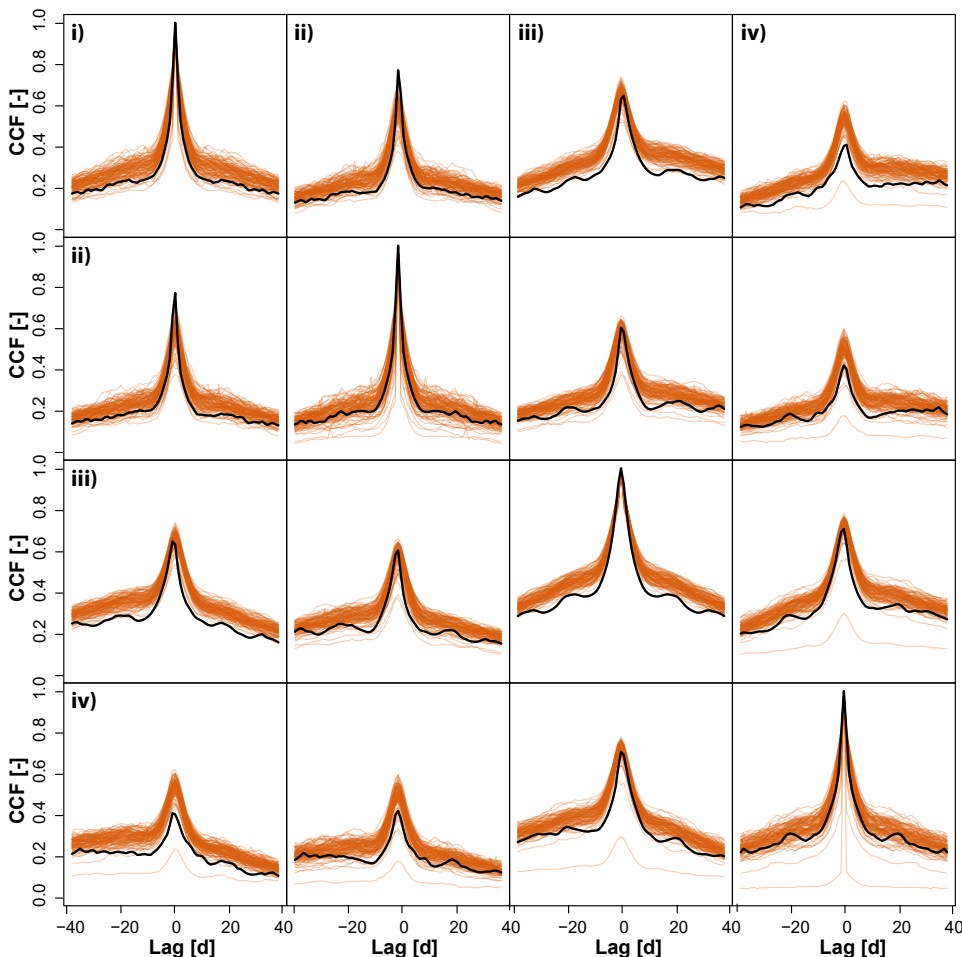

**Figure 7.** Comparison of observed (black) and simulated (orange) cross-correlation functions (ccfs) for the daily discharge values for the four catchments (i–iv) in the Pacific Northwest.





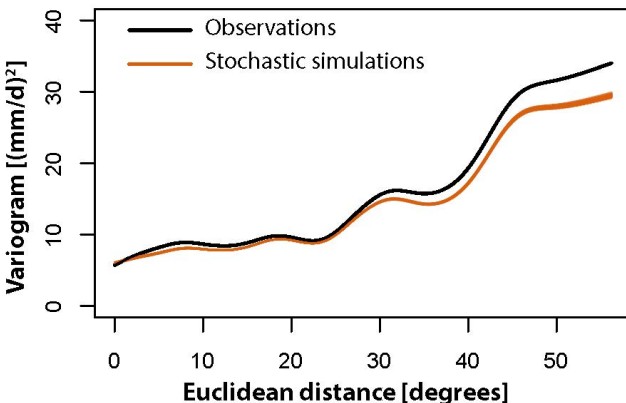

**Figure 8.** Comparison of observed (black) and simulated (orange) variograms.

Spatial dependencies are not only maintained for the bulk of the distribution, by which we mean the part of the distribution excluding extremes or outliers, but also for extreme values as illustrated by the peak-over-threshold (POT) values for the different stations in the three illustration regions (Fig. 9). These results show that besides regional flood co-occurrences, the temporal clustering behavior of events is also reproduced.





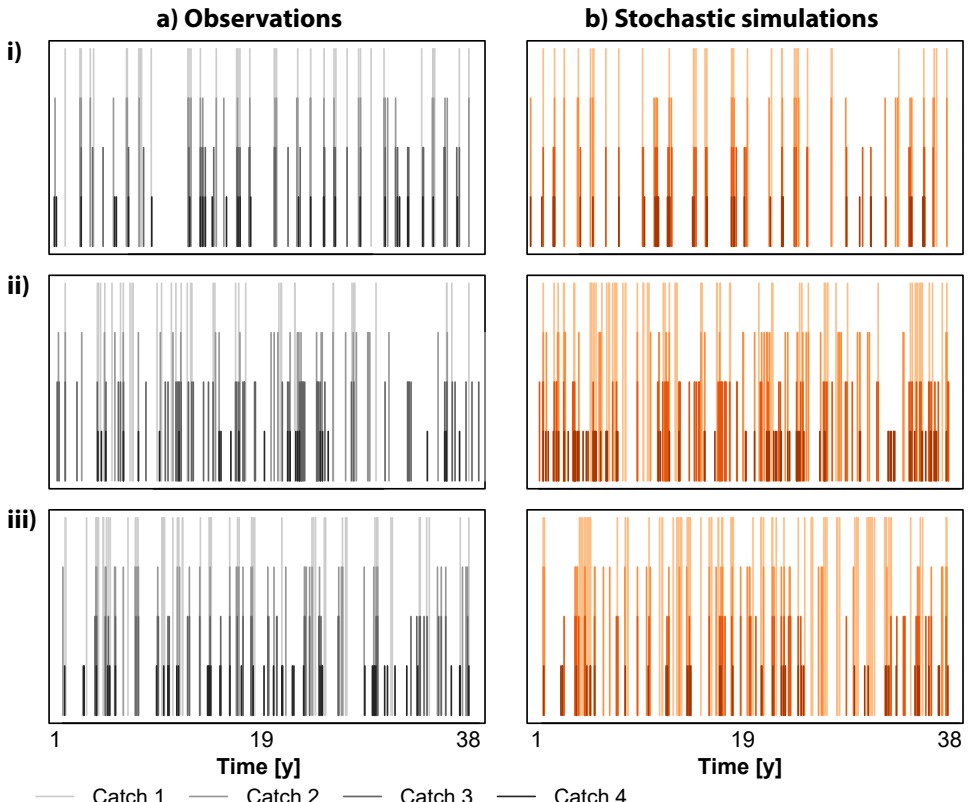

**Figure 9.** Observed (a; black) vs. stochastically simulated (b; orange) POT events for the four stations (color shadings) in the three regions
(i) Pacific Northeast, (ii) Texas, and (iii) Mid-Atlantic. Vertical bars indicate event occurrences over the 38 years.

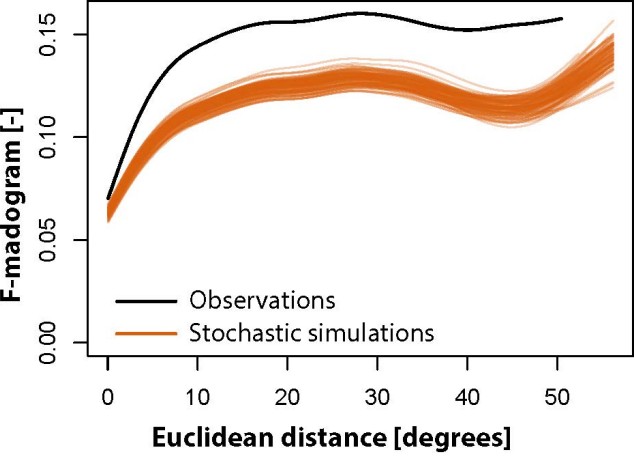

**Figure 10.** Observed (black) vs. simulated (orange) F-madograms plotted against Euclidean distance. The lower the value, the higher the
dependence between a pair of stations.





The F-madograms shown in Fig. 10 indicate that there is generally good agreement between observed and simulated spatial dependence despite a slight overestimation of the spatial dependence of floods by the stochastic simulations. This overestimation means that a certain pair of stations may experience more joint floods according to the simulations than seen in the observations. The good agreement between observed and simulated spatial dependence and the weak overestimation in spatial dependence is also visible if we look at the tail dependence coefficient $\chi$ for the two thresholds 0.8 and 0.95 (Fig. 11). For most pairs of stations, both the simulations and observations indicate no tail dependence (grey dots). If the observations indicate tail dependence, the simulations mostly also simulate upper tail dependence (green dots). Only in very few cases, the simulations do not capture the tail dependence in the observations (black dots). There are, however, quite a few cases where the simulations indicate tail dependence despite its absence in the observations (orange dots). Overall, the model shows good performance in the reproduction of observed spatial (in)dependencies.

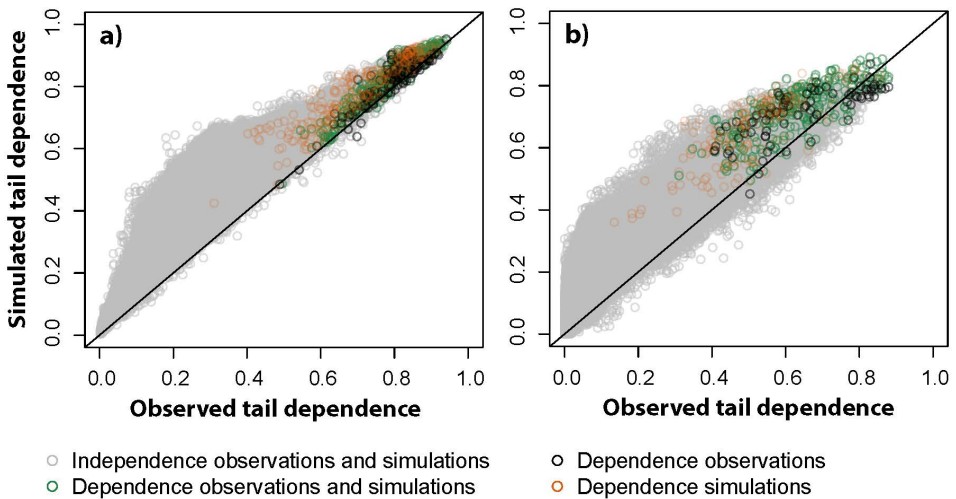

**Figure 11.** Observed vs. simulated tail dependence coefficient $\chi$ for the two thresholds 0.8 (a) and 0.95 (b). Grey dots indicate pairs of stations with no upper tail dependence, green dots pairs where both observations and simulations indicate upper tail dependence, black dots pairs of stations where only the observations indicate tail dependence, and orange dots pairs of stations where only the simulations indicate tail dependence.

## 5  Discussion

Similar to the Fourier transform based simulation approach *PRSim* by Brunner et al. (2019), the wavelet-based simulation approach *PRSim.wave* presented here allows for the generation of many time series of the same length as the observed series. This means that the representation of temporal dependence is limited to ranges within the length of the observed series. However, the modeled range of dependence is also limited to the one in the observed series if one very long time series is generated. In addition to the representation of temporal dependence, *PRSim.wave* allows for the reproduction of realistic



spatial dependencies both in the general distribution and in extreme events. This representation of spatial dependence is not possible if using the Fourier transform as in *PRSim*. This difference between methods may be related to the fact that the wavelet transform compared to the Fourier transform does not necessitate a transformation to the normal domain, and a back transformation to the domain of the skewed distribution, which has been shown to weaken spatial correlations (Embrechts et al., 2010; Papalexiou, 2018). The use of the kappa distribution in combination with the spatio-temporal model allows for the generation of extremes beyond the range of the observed values. However, it requires the fitting of many parameters, which make the model non-parsimonious (Koutsoyiannis, 2016). Depending on the application, the approach can therefore be used with a distribution with fewer parameters, a distribution fitted to a monthly instead of daily scale, or the empirical distribution.

The application of the approach is not limited to observed streamflow time series. It is applicable to other variables such as precipitation if combined with a suitable distribution as well as to modeled time series. The use of streamflow time series generated with a hydrological model extends the application of *PRSim.wave* to climate impact studies where a hydrological model is driven by meteorological time series generated with global and/or regional climate models.

## 6  Conclusions

Our results show that the continuous, wavelet-based stochastic simulation approach *PRSim.wave* reliably simulates discharge and extremes at multiple sites. Thanks to a spatio-temporal model based on phase randomization, temporal short- and long range dependencies, non-stationarities, and spatial dependencies are reproduced. In combination with the parametric kappa distribution, spatial extremes at multiple sites can be reliably simulated as well. The stochastic approach of *PRSim.wave* is very flexible and easy to use because of its availability in the R-package *PRSim*. Its versatility and advantageous properties make it generally useful for various water management applications where spatio-temporal patterns are of interest and, in particular, valuable for hazard assessments requiring information on spatial extremes.

*Code and data availability.*  The wavelet-based stochastic simulation procedure for multiple sites *PRSim.wave* using the empirical, kappa, or any other distribution and some of the functions used to generate the validation plots are provided in the R-package *PRSim*. The stable version can be found in the CRAN repository https://cran.r-project.org/web/packages/PRSim/index.html, and the current development version is available at https://git.math.uzh.ch/reinhard.furrer/PRSim-devel. The observational discharge data was provided by the USGS and can be downloaded via: https://waterdata.usgs.gov/nwis.

*Author contributions.*  MIB developed, set up, evaluated, and implemented the stochastic simulation approach in the R-package *PRSim*. MIB wrote the first draft of the manuscript and produced the figures shown therein. EG provided input on the evaluation statistics and reviewed and edited the manuscript.





*Competing interests.* The authors declare that they have no conflict of interest.

*Acknowledgements.* We thank Balaji Rajagopalan and Christopher Torrence for valuable discussions, which helped to shape the simulation approach. This work was supported by the Swiss National Science Foundation via a PostDoc.Mobility grant (Number: P400P2_183844, granted to MIB). Support for EG was provided by the Regional Climate Uncertainty Program (RCUP), an NSF-supported program at NCAR.





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
