# Peer review of "Stochastic simulation of streamflow and spatial extremes: a continuous, wavelet-based approach."

_Hydrology and Earth System Sciences, 2019_

## Referee Comment (RC1) · Anonymous Referee #1 · 29 Jan 2020

GENERAL COMMENTS The manuscript proposes a wavelet-based phase randomization approach to generate multisite hydrological extremes, which is argued to be able to capture both spatial dependencies and non-stationarities. The stochastic simulated data is reconstructed with the same values as original data in another temporal order, and this assures the reproduction of temporal dependence, extremes and non-stationarities. In addition, the multiplication of the same set of random phases to all the investigated sites ensures spatial dependences. The paper is clearly written, and the idea is well explained. However, I have listed some comments below which needs to be addressed before the manuscript is considered further for publications.

Major comments: COMMENT1: The spatial dependence can be simulated by using the same randomized phases for multiple time series among sites. You claimed this in your introduction and Section 3.2 step 1 without detailed explanation and references. From my understanding, the phases represent the time of the timing of changes or variations (in your case, it is extreme events), so if you use the same randomized phases, it is expected that there will be spatial dependence of extremes among the sites you investigated. Thus, could you give explanations or illustrations on this statement.

COMMENT2: Section 3.2, step 1 shows how the random phases are computed from a random discharge time series from a normal distribution with mean 0 and standard deviation 1. Could you explain the specific reason to choose normal distribution or have you tested with other distribution for example, gamma and kappa distribution? Another similar question is about the selection of wavelet family and scale. In this study, you use Morlet wavelet, what is the specific reason to use this wavelet family, and how about other wavelet families, e.g., Paul, DOG and Marr (Torrence and Compo, 1998)? The sensitivity of your approach to the selection of wavelet family and scale? For the application of wavelet method in real world, the selection of wavelet family and scales is of great importance.

Additionally, I am unable to visualise where exactly the random phases go back into equation 3? What should have been the values of these phases with the observed data in the first place? Why should these phases be the same? I suspect the only impact these phases have is on spatial dependence. In that case why are the majority of results that are presented focussing on temporal dependence attributes? Figure 6 gives some flavour of the multi-suite stochastic generation. It looks troublingly similar to the observations. What would happen if non-identical phases used across all the sites? These questions are important to address to establish the contribution here is an improved representation of spatial dependence compared to other alternatives. Regarding other alternatives, is there a possibility of comparing this method with the commonly used spatial stochastic generators such as SPIGOT or other equally worthy

alternatives? I would like to see the variogram figure possibly expanded to also show alternate outcomes using other formulations of spatial dependence (ie different ways of specifying phases).

COMMENT3: Section 3.2, step 4 mentions how the reconstruction is done by using the inverse wavelet transform (Eq. (3)) combining the derived random phases and the amplitudes in previous steps. Could you write the combination explicitly in the from of the equation (i.e., include the random phases in the Eq. (3))?

Minor comments: Line 147: h=100 wavelet scales, I think you mean number of wavelet scales is 100, not the scale itself equals 100. Wavelet scales should be hj = h0*2^(j*dj), j=0,1,...J. Or is this some parameter that is specific to a continuous wavelet transform? Why is it of relevance, what impact does it make, and why use the same value for all locations is something that should be discussed.  TORRENCE, C. & COMPO, G. P. 1998. A Practical Guide to Wavelet Analysis. Bulletin of the American Meteorological Society, 79, 61-78.

---

## Short Comment (SC1) · 19 Feb 2020

Ioannis Tsoukalas (e-mail: itsoukal@mail.ntua.gr)

**Introduction**

This is an interesting work aiming to provide a method (as well as implement it in an R package; function called PRSim.wave within PRsim R package) for the simulation of multivariate hydrological processes (for now, focusing on streamflow) - which according to the results presented by the Authors has a good potential, requiring yet some improvements.

In general, I find the manuscript well-organized and straightforward to understand, yet in my view, there are several points that require the Authors attention.

All comments and suggestions are meant to be constructive and aim to improve the quality of the manuscript, as well as the findings obtained.

**Comments**

**L6-7.** I suggest to write: "To do so, we propose the stochastic simulation approach called Phase Randomization Simulation using wavelets (here after called *PRSim.wave*) which combines… ".

**L11.** To avoid confusing the reader, and provided that a few lines above it is mentioned that "We apply and evaluate *PRSim.wave* on a large set of 671 catchments in the contiguous United States.", I suggest to write: "…at multiple sites (up to four)…"

**L34.** I wonder what are the "potential non-stationarities" mentioned by the Authors? Are you refereeing to the typical cyclostationary behavior exhibited by hydrological processes?

Given the opportunity, and as a side note, I would like highlight that stationarity is an essential tool for inferencing from data (e.g., model fitting). Stationarity should not be seen as a shortcoming, nor dead. Non-stationarity implies non-ergodicity, which in turn makes inference from observed data impossible, unless of course the deterministic dynamics of the process are known; which in my understanding, is never the case in hydrological sciences. On this topic, I recommend the recent work of Serinaldi et al. (2018), with emphasis on section 4.2, as well as the works of Koutsoyiannis and Montanari, (2007), (2015), Lins and Cohn (2011), Matalas (Matalas, 2012), and Montanari and Koutsoyiannis (2014), that argue in favor of stationarity. See also the very interesting, note of Harry F. Lins[1]. As Harry F. Lins concludes his note:

Stationarity ≠ static

Non-stationarity ≠ change (or trend)

**L36.** I am not sure what is the meaning of "continuous" here? Can you please elaborate/specify? Also, some references would be useful.
* * *
[1] http://www.wmo.int/pages/prog/hwrp/chy/chy14/documents/ms/Stationarity_and_Nonstationarity.pdf

**L40.** Although I understand its rationale, I am not a big fan of the now-typical classification of stochastic models on "parametric and non-parametric", since in my view, there is no model without parameters. Typically, the literature uses the term "non-parametric" to refer to approaches that use some kind of resampling mechanism (e.g., k-nn algorithm) and/or "non-parametric" distribution functions (e.g., kernel-based approximation of the density function) to generate synthetic data. But, one should take a moment and think, are these really non-parametric? Isn't the $k$ (i.e., the number of nearest neighbors) in k-nn algorithm a parameter? Isn't the choice of the kernel smoothing function (e.g., normal, epanechnikov, box, triangle) a parameter? Isn't the bandwidth of the kernel a parameter (also called the smoothing parameter)? Aren't the data *per se* used as parameters (e.g., when a non-parametric method relies on the sampling from the empirical CDF or kernel-based CDF. What if we have new data or alter a few? Does the model change?)? Having said these, I suggest to the Authors to reconsider using the employed classification, as well as review the recent literature (e.g., Serinaldi and Kilsby, 2014; Tsoukalas et al., 2019) for finding an alternative classification.

**L41-43.** The Authors write: "…and temporal disaggregation models such as fractional Gaussian noise models (Mandelbrot, 1965), fast fractional Gaussian noise models (Mandelbrot, 1971), broken line models (Mejia et al., 1972), and fractional autoregressive integrated moving average models (Hosking, 1984).".

To clarify, these are not disaggregation models, but models able to simulate processes exhibiting long-range dependence (particularly, designed to simulate fractional Gaussian noise (fGn) processes or else, processes exhibiting Hurst behavior). See a similar discussion in the introduction section of Tsoukalas et al. (2018b).

**L44-45.** The Authors write: "Nonparametric models are based on disaggregation and resample from the data with perturbations and include…".

I think that this statement can be confusing, and needs some refinement. "Non-parametric" models are not necessarily based on the notion of disaggregation. Of course, the literature offers "non-parametric" disaggregation methods (e.g., Lee et al., 2010; Tarboton et al., 1998), but this does not makes all "non-parametric" methods, methods that "based on disaggregation and resample". Further details on disaggregation methods can be found on the seminal works of Valencia and Schakke (1973), and Mejia and Rousselle (1976), as well in the work of Koutsoyiannis (2001) who provide a detailed overview on the subject. For a more recent overview and discussion on the topic of disaggregation and multi-temporal simulation see also work of Tsoukalas et al. (2019).

**L48-49.** The Authors write: "…but none of these time domain methods can capture the spectral properties of the observed time series (Erkyihun et al., 2017).".

In my view this statement is a bit confusing, requiring the Authors attention, for two reasons.

> 1) A timeseries (i.e., a sequence of observations ordered in time) does not has spectral properties, it exhibits some form of dependence structure (which can be quantified using statistics/stochastics, e.g., through the empirical correlation coefficients and the empirical spectrum). What has spectral properties is the stochastic process that it is assumed that generated the observed timeseries.

> 2) Having said the above, and since correlation and spectrum are interrelated quantities, if a model is capable of reproducing the process's correlation structure it also reproduces its

spectrum (and vice versa). For further details and references, see my previous comment (Tsoukalas, 2019) on a recent work co-authored by the first Author of this work.

**L49-50.** In my view the sentence "Furthermore, these time-domain models struggle with the representation of spatial dependence" is a bit "strict", since as far as I see it, there is no struggle, but many research efforts (past, and new).

The stochastic hydrology literature offers several "time-domain" models that can simulate parsimoniously multivariate processes, including both stationary and cyclostastionary processes (e.g., Efstratiadis et al., 2014; Koutsoyiannis, 2001, 2000), reproducing also the moments of the observed processes (typically up to third order). Further to these models/methods, more recent approaches allows the parsimonious simulation of multivariate stationary and cyclostationary processes with any marginal distribution and correlation structure (Kossieris et al., 2019; Tsoukalas, 2018; Tsoukalas et al., 2018a, 2018b), also in a multi-scale context (Tsoukalas et al., 2019). Apart from the last work, for another multi-scale and multivariate simulation study involving daily rainfall at 4 sites the Authors are referred to Appendix D, section D.2, of Tsoukalas (2018). Therefore, taking into consideration the above-mentioned works I would suggest the Authors to revise the sentence accordingly, as well as provide some references.

**L54-55.** The Authors write: "In contrast to time-domain models, frequency-domain models allow for the simulation of surrogate data with the same Fourier spectra as the raw data".

This can be also true for time-domain methods (but not a good modelling practice in either of the two cases; see below). For instance, if one employs an AR or MA model of high order can simulate a realization of a process exhibiting exactly the empirical autocorrelation coefficients up to the order dictated by the model. However, this is not a good modeling practice since it is well-known that the empirical estimators of auto- (and cross-) correlation coefficients are (downward) biased (Beran, 1994; Koutsoyiannis, 2003, 2000), especially in the case of long-range dependence, short samples, and large lags. See also Matalas (1967 p. 945) who remark that:

> "*Parameters that are determined in terms of high order moments of large time lags are subject to large standard errors and consequently large operational biases. Operational biases can never be eliminated, but they can be minimized by the use of regionalization to account for the temporal and spatial variations inherent in the historic sequences...*".

Of course, the same applies for the empirical estimators of spectrum (see the comparative work of (Dimitriadis and Koutsoyiannis (2015)). Note that this kind of approaches are not parsimonious (since all the empirical estimates used in model fitting are essentially model parameters). To cope with these, the recent literature (Kossieris et al., 2019; Tsoukalas et al., 2019, 2018b), as well some works already cited in the manuscript (i.e., Papalexiou (2018)), has leaned towards the use of parametric models (e.g., with two or three parameters) to parsimoniously describe the dependence structure of the processes. The Authors are referred to the work of Koutsoyiannis (2000) which in my view popularized that idea in hydrological domain, also introducing a parsimonious two-parameter auto-correlation structure. It is also interesting to note the work of Papalexiou (2018) (already cited in the manuscript), who employed the functional form provided by the survival function of a distribution to define several auto-correlation structures.

**L70-72.** The Authors write: "In addition, it may help to improve the representation of spatial dependencies because it does not require a transformation to the normal distribution and back to

the original, skewed distribution, which usually weakens spatial correlations (Embrechts et al., 2010).“

First, the comment on “weaken spatial correlations” applies for all “types” of correlations (that emerge from the mapping/transformation from the Gaussian to the actual domain) – not only spatial. Particularly, in the case of stochastic processes, it also applies for the auto-correlation structure of a stationary processes, as well as for the season-to-season correlations of a cyclostationary process (Tsoukalas et al., 2018a, 2017). It also holds for multivariate cases. However, I am afraid that I cannot see the improvement of the representation of spatial dependencies mentioned above by the Authors. The cross-correlations as well as the auto-correlation are still not accurately reproduced (see my comments below on the results/plots). It is my understanding that a previous comment of mine (Tsoukalas, 2019) on a recent work co-authored by the first Author of this work holds also for this method. This is due to the following:

> **L145.** The Authors write: “Derivation of random phases: A random discharge time series (white noise) of the same length as the input series is sampled from a normal distribution with mean 0 and standard deviation 1.”

> **L170-173.** The Authors write: “Transformation to kappa distribution: The simulated values are transformed to the kappa domain using the fitted daily kappa distributions from Step 2. For each day, a random sample is generated from the fitted, daily kappa distribution. The simulated values are replaced by the values generated from the kappa distribution using rank-ordering. This procedure is repeated for each day in the year.”

Based on the above the method presented herein depends on an auxiliary Gaussian process and uses the target ICDFs, as well as the rank-correlations to establish the (auto- and cross-) dependence structure. It is reminded that such a procedure will preserve the ranks correlation coefficients (which do not depend on the marginals) but not the Pearson’s, (which depends on the marginals; since it involves the cross-product moment of the among the variables). For further details the Authors are referred to the comment mentioned above, as well as in the references therein. It is my understanding that the mechanics of the method that dictate the preservation of ranks is the reason why the auto- (and cross-) correlations are not so well reproduced by the proposed method.

**L149 (and elsewhere)**: The use of Kappa distribution. As mentioned in a previous comment of mine in HESS related with a work co-authored by an Author of this manuscript there are few complications worth considering when using the Kappa distribution. The following comments are excerpted with minimum or no modifications at all from Tsoukalas (Tsoukalas, 2019).

1. Since you are using the Kappa distribution it could be insightful to mention that under certain parameter combinations, this distribution may lead to infinite moments. This can be a delicate issue, since if the fitted distribution exhibits infinite variance then the Pearson’s correlation cannot be defined (the denominator contains the variance), and thus the proposed model (as well as many other models) cannot be used. This situation is discussed in section 3.4 of Tsoukalas et al. (2018b; and references therein), where it is advocated (based on empirical, as well as theoretical reasoning) that physical processes are characterized by finite variance (Koutsoyiannis, 2016).
   Particularly, if $X$ is a Kappa-distributed random variable, and $\mu_r = E[X^r]$ denotes the $r^{th}$ raw moment, as discussed in Hosking (1994), and elsewhere, the existence of the $r^{th}$ depends on the values of $h$ and $k$. Specifically, the moments exist:

$$\text{for all } r \quad \text{if} \quad h \geq 0 \text{ and } k \geq 0$$
$$\text{for } r < -1/hk \quad \text{if} \quad h < 0 \text{ and } k \geq 0, \text{and}$$
$$\text{for } r < -1/k \text{ if } k < 0$$

It is also interesting to mention that Hosking (1994) notes that the first four moments cannot uniquely determine the parameters of the distribution, since some combinations of moments (expressed by skewness and kurtosis coefficients) correspond to different pairs of $h$ and $k$.

2. How do you handle negative values? As far as I am aware the left (and right) support of Kappa distribution is not necessarily zero (e.g., when $k = 0$ and $h \leq 0$, then the supports of the distribution are, $-\infty < x < \infty$; see Hosking (1994)). In any case, the generation of negative values can be eliminated by using a distribution function defined in the positive real line. Particularly, I would suggest the investigation/use of the Generalized Gamma and Burr type-XII distributions, which are more parsimonious (they entail three parameters; instead of four as in Kappa) and were found adequate for modelling of hydrometeorological variables; particularly rainfall (e.g., Papalexiou and Koutsoyiannis, 2016). Examples of their use within the context of stochastic modelling can be found the work Papalexiou (2018), as well as in Tsoukalas et al. (2019, 2018b) and Tsoukalas (2018).

**L158-160.** The Authors write: "We fit a separate distribution for each day to take into account seasonal differences in the distribution of daily streamflow values. To do so, we use the daily values in a 30-day window around the day of interest."

Can you please elaborate on this? Just to be sure, for each "site" and for each day of the year a Kappa distribution has been fitted with different parameters? If this is the case, just for the marginal behavior, and for each site you fitted Kappa 365 times, which implies that the model has $365 \times 4$ (the number of parameter of Kappa) $= 1460 \times$ the number of sites, parameters (not accounting those for the specification of the auto and cross-dependence structure of the process). If this is the case, I am afraid that this is not a parsimonious model, something that should be clearly stated in the manuscript (also mentioning the total number of its parameters).

Also, I don't think that it is reasonable to assume that days belonging in the same month (e.g., the 19th and the 20th of August) have different marginal distribution (although, I have seen stochastic simulation related works following that approach, I am not aware of any paper supporting this assumption). The standard approach for daily (or finer time) scales is to consider stationarity within the monthly interval (i.e., in the case of daily data consider that all days belonging in the month have the same marginal distribution). An arguably more parsimonious approach, since in this case the total number of parameters for the marginal behavior would be $12 \times 4$ (the number of parameter of Kappa) $= 48 \times$ the number of sites. Note that the number of parameters could be further reduced by using, instead of Kappa, alternative 2- or 3-parameter distribution models.

**L162-164.** The Authors write: "In a few regions with many zero discharge values (e.g. some catchments in the Great Plains) fitting the kappa distribution is not possible and we therefore use the empirical distribution instead."

This is a work-around that could work, but I wonder, why not use an alternative distribution model (e.g., zero-inflated or mixed) that can model simultaneously both the discrete (i.e., probability of no discharge) and continuous part (i.e., distribution of non-zero discharge) of the process? Also, can you

provide an estimate on the number of cases where the empirical distribution is employed instead of Kappa?

**L218-219.** The Authors write: "the seasonal (3d) and monthly distributions (3e–g) are well captured by the simulations."

This is a confusing description of the plots. The (3e–g) plots show that some seasonal summary statistics are reproduced (i.e., monthly mean, monthly maxima, monthly minima), not the seasonal distributions (to do so you need alternative plots, comparing the empirical distribution of each month with the corresponding theoretical one). Also, I don't understand what plot 3d shows? What does "seasonal statistics" means in the title of this plot? What does the phrase "the seasonal discharge distribution" means at the legend of Figure 3? These are box-plots, and by no means should be used to compare distribution functions (they provide way too few information - about specific quantiles).

**L220-222.** The Authors write: "the temporal correlation characteristics (4a–c), …. are well captured by the simulations as well."

In my view there is an important difference between the simulated and empirical autocorrelation coefficients. For instance, by eye-balling the median of the orange lines (simulated) for time lag 10 we get a value about equal to 0.5, while the observed one is 0.4. This should be clearly stated in the manuscript. For the readers convenience, I also suggest the inclusion of a line depicting the median of the simulated quantities (orange lines). Providing lines for a low and high quantile (say 0.05 and 0.95) would be nice also. This comment applies for all similar plots throughout this manuscript.

**L223-225.** The Authors write: "Both high- and low- extremes are realistically modeled as illustrated by the distributions of the above and below threshold events of the four catchments in the Pacific Northwest (Fig. 5)."

To avoid confusion with actual distribution functions, e.g., the Kappa, instead of using the phase "as illustrated by the distributions of the above and below threshold events", I would suggest the use of phrase similar to: "as illustrated box-plots of Fig.5, constructed by the values of the above and below threshold events".

**L223-225.** Please remind the reader the selected threshold values.

**Figure 6 (legend).** I assume that you wanted to write: "… (b) for the **three** catchments in the….".

**L230-233.** The Authors write: "This visual impression of a good performance with respect to the reproduction of spatial correlations in daily discharge data is confirmed by comparing observed and stochastically simulated cross-correlation functions for the catchments in the Pacific Northwest (Fig. 7). Both the shape and magnitude of the cross-correlation functions are well simulated".

I am afraid that this sentence is needs some refinement, since as with the case of auto-correlation coefficients, Fig. 7 shows a significant deviation of the simulated cross-correlation coefficients from the observed ones. This should be reflected in the manuscript. For the readers convenience, I would also suggest the inclusion of a line depicting the median of all simulations (orange lines). Providing lines for a low and high quantile (say 0.05 and 0.95) would be nice also.

**Figure 7.** There is something wrong with the labeling of the panels (i.e., multiple panels are labeled as ii, iii, iv, while some others are completely unlabeled).

**L239.** The Authors write: "…but also for extreme values as illustrated by the peak-over-threshold (POT) values for the different stations in the three illustration regions (Fig. 9). These results show that besides regional flood co-occurrences, the temporal clustering behavior of events is also reproduced."

In my view, Figure 9 is not very informative (the y-axis has been omitted intentionally?). Can you please provide an alternative figure, as well as a quantitative metric, quantifying the reproduction of temporal clustering behavior of events?

**Discussion section.** In my view all the above points should be discussed in this section, highlighting also the limitations of the presented method. Further to these, it should be noted that the proposed model has been tested for multivariate problems involving 4 processes, as well as the method is capable of generating synthetic timeseries with length equal to the observed one (I haven't read how to handle the case where one wants to generate longer timeseries – is it possible to generate synthetic timeseries with length different than the observed one?).

**L258-259.** The Authors write: "This difference between methods may be related to the fact that the wavelet transform compared to the Fourier transform does not necessitate a transformation to the normal domain, and a back transformation to the domain of the skewed distribution, which has been shown to weaken spatial correlations".

I think that the answer is is much simpler than the one stated above. The former method is simply designed for the simulation of univariate processes, i.e., not to account for the cross-correlations (or cross-spectrum) among processes. I suggest the Authors to consider more carefully the "mechanics" of the aforementioned methods, and revise the sentence accordingly. Also, as mentioned before, the comment on "weaken spatial correlations" applies for all "types" of correlations (that emerge from the mapping/transformation from the Gaussian to the actual domain) – not only spatial. Particularly, in the case of stochastic processes, it also applies for the auto-correlation structure of a stationary processes, as well as for the season-to-season correlations of a cyclostationary process (Tsoukalas et al., 2018a, 2017). It also holds for multivariate cases.

**L270-271.** The Authors write: "Thanks to a spatio-temporal model based on phase randomization, temporal short- and long range dependencies, non-stationarities, and spatial dependencies are reproduced."

Please consider my comments on the parameterization (i.e., number of parameters), as well as on the performance of the model and revise this sentence accordingly.

**A general comment.** A final comment regards the title of the manuscript, which is: "Stochastic simulation of streamflow and spatial extremes: a continuous, wavelet-based approach". By reading the paper I see that the Authors pay special focus on the reproduction of extremes, but it is not clear to me why this model is different from any other in that aspect (e.g., see those mentioned above)? What makes this model suitable when aiming to reproduce extremes? Other similarly parameterized models behave differently? If yes, why? I believe that a theoretical justification or even an empirical comparison with alternative model(s) would be particularly useful and an added value for the paper.

Regards,

Ioannis Tsoukalas

**References**

Beran, J., 1994. Statistics for long-memory processes. CRC press.

Dimitriadis, P., Koutsoyiannis, D., 2015. Climacogram versus autocovariance and power spectrum in stochastic modelling for Markovian and Hurst–Kolmogorov processes. Stoch. Environ. Res. Risk Assess. 29, 1649–1669. https://doi.org/10.1007/s00477-015-1023-7

Efstratiadis, A., Dialynas, Y.G., Kozanis, S., Koutsoyiannis, D., 2014. A multivariate stochastic model for the generation of synthetic time series at multiple time scales reproducing long-term persistence. Environ. Model. Softw. 62, 139–152. https://doi.org/10.1016/j.envsoft.2014.08.017

Hosking, J.R.M., 1994. The four-parameter kappa distribution. IBM J. Res. Dev. 38, 251–258. https://doi.org/10.1147/rd.383.0251

Kossieris, P., Tsoukalas, I., Makropoulos, C., Savic, D., 2019. Simulating Marginal and Dependence Behaviour of Water Demand Processes at Any Fine Time Scale. Water 11, 885. https://doi.org/10.3390/w11050885

Koutsoyiannis, D., 2016. Generic and parsimonious stochastic modelling for hydrology and beyond. Hydrol. Sci. J. 61, 225–244. https://doi.org/10.1080/02626667.2015.1016950

Koutsoyiannis, D., 2003. Climate change, the Hurst phenomenon, and hydrological statistics. Hydrol. Sci. J. 48, 3–24. https://doi.org/10.1623/hysj.48.1.3.43481

Koutsoyiannis, D., 2001. Coupling stochastic models of different timescales. Water Resour. Res. 37, 379–391. https://doi.org/10.1029/2000WR900200

Koutsoyiannis, D., 2000. A generalized mathematical framework for stochastic simulation and forecast of hydrologic time series. Water Resour. Res. 36, 1519–1533. https://doi.org/10.1029/2000WR900044

Koutsoyiannis, D., Montanari, A., 2015. Negligent killing of scientific concepts: the stationarity case. Hydrol. Sci. J. 60, 1174–1183. https://doi.org/10.1080/02626667.2014.959959

Koutsoyiannis, D., Montanari, A., 2007. Statistical analysis of hydroclimatic time series: Uncertainty and insights. Water Resour. Res. 43, 1–9. https://doi.org/10.1029/2006WR005592

Lee, T., Salas, J.D., Prairie, J., 2010. An enhanced nonparametric streamflow disaggregation model with genetic algorithm. Water Resour. Res. 46, 1–14. https://doi.org/10.1029/2009WR007761

Lins, H.F., Cohn, T.A., 2011. Stationarity: Wanted dead or alive? J. Am. Water Resour. Assoc. https://doi.org/10.1111/j.1752-1688.2011.00542.x

Matalas, N.C., 2012. Comment on the Announced Death of Stationarity. J. Water Resour. Plan. Manag. 138, 311–312. https://doi.org/10.1061/(ASCE)WR.1943-5452.0000215

Matalas, N.C., 1967. Mathematical assessment of synthetic hydrology. Water Resour. Res. 3, 937–945. https://doi.org/10.1029/WR003i004p00937

Mejia, J.M., Rousselle, J., 1976. Disaggregation models in hydrology revisited. Water Resour. Res. 12, 185–186. https://doi.org/10.1029/WR012i002p00185

Montanari, A., Koutsoyiannis, D., 2014. Modeling and mitigating natural hazards: Stationarity is immortal! Water Resour. Res. 50, 9748–9756. https://doi.org/10.1002/2014WR016092

Papalexiou, S.M., Koutsoyiannis, D., 2016. A global survey on the seasonal variation of the marginal distribution of daily precipitation. Adv. Water Resour. 94, 131–145. https://doi.org/10.1016/j.advwatres.2016.05.005

Serinaldi, F., Kilsby, C.G., 2014. Simulating daily rainfall fields over large areas for collective risk estimation. J. Hydrol. 512, 285–302. https://doi.org/10.1016/j.jhydrol.2014.02.043

Serinaldi, F., Kilsby, C.G., Lombardo, F., 2018. Untenable nonstationarity: An assessment of the fitness for purpose of trend tests in hydrology. Adv. Water Resour. https://doi.org/10.1016/j.advwatres.2017.10.015

Tarboton, D.G., Sharma, A., Lall, U., 1998. Disaggregation procedures for stochastic hydrology based on nonparametric density estimation. Water Resour. Res. 34, 107. https://doi.org/10.1029/97WR02429

Tsoukalas, I., 2019. Interactive comment on "Technical note: Stochastic simulation of streamflow time series using phase randomization" by Manuela I. Brunner et al.

Tsoukalas, I., 2018. Modelling and simulation of non-Gaussian stochastic processes for optimization of water-systems under uncertainty. PhD Thesis, Department of Water Resources and Environmental Engineering, National Technical University of Athens (Defence date: 20 December 2018).

Tsoukalas, I., Efstratiadis, A., Makropoulos, C., 2019. Building a puzzle to solve a riddle: A multi-scale disaggregation approach for multivariate stochastic processes with any marginal distribution and correlation structure. J. Hydrol. 575, 354–380. https://doi.org/10.1016/j.jhydrol.2019.05.017

Tsoukalas, I., Efstratiadis, A., Makropoulos, C., 2018a. Stochastic Periodic Autoregressive to Anything (SPARTA): Modeling and simulation of cyclostationary processes with arbitrary marginal distributions. Water Resour. Res. 54, 161–185. https://doi.org/10.1002/2017WR021394

Tsoukalas, I., Efstratiadis, A., Makropoulos, C., 2017. Stochastic simulation of periodic processes with arbitrary marginal distributions, in: 15th International Conference on Environmental Science and Technology. CEST 2017. Rhodes, Greece.

Tsoukalas, I., Makropoulos, C., Koutsoyiannis, D., 2018b. Simulation of stochastic processes exhibiting any-range dependence and arbitrary marginal distributions. Water Resour. Res. https://doi.org/10.1029/2017WR022462

Valencia, R.D. V., Schakke, J.C., 1973. Disaggregation processes in stochastic hydrology. Water Resour. Res. 9, 580–585. https://doi.org/10.1029/WR009i003p00580

---

## Referee Comment (RC2) · Sandhya Patidar (Referee) · 26 Feb 2020

Title: Stochastic simulation of streamflow and spatial extremes: a continuous, wavelet-based approach (Brunner et al)

Review report by Dr Sandhya Patidar

The paper presents a continuous wavelet-based phase randomisation approach for the stochastic generation of streamflow time series. This is an interesting paper that substantially extends the original techniques developed in [1] for stochastic simulation of streamflow using Fourier transformation based phase randomisation. The original

method, presented in [1] is associated with certain limitations such as application of Fourier transformation does not account for the non-stationarities in time series and resulted in an underestimation of spatial dependencies (e.g. cross-correlation) in both daily discharge and extreme events across the multiple sites. The present paper address some of the limitations observed in [1] by replacing Fourier transformation with a complex wavelet-based approach. The efficiency of original method has been evaluated to generate realistic series for distributional and temporal correlation characteristics and is validate through the application across four catchments in Switzerland. The proposed model is applied to a large dataset of 671 catchments in the contiguous United States and the efficiency of the model has been evaluated by assessing its ability to capture distributional and temporal characteristics at individual sites along with the spatial dependencies across the multiple sites, and extreme events (floods) including duration and volume.

Some general comments – Section 1 - Considering the theme of paper that signifies the application of wavelet approach, the Introduction section presents an interesting critical review on recently developed/applied modelling schematics that involves wavelet-based phase randomisation as a key approach. Line 70-73: To add clarity it would be helpful if the authors' team add some brief explanation on how data normalisation procedure and back transformation impacts the spatial dependencies. Line 87-88: Please add some clarity on how continuous wavelet transform is more effective than a discrete wavelet transforms in minimising/overcoming issues around the long-term periodicities and/or non-stationarities. Is there any specific studies carried out to investigate such issues.

Section 2 - Theoretical background section provides suffiecient details on the wavelet decomposition approach. Section 3 Data – For illustration and validation purposes, dataset are organised in three different region based on the general hydrological characteristics. It is not clear which specific properties has been used and how rigorously they have been appied. I think, this work could have benefitted if Authors' have considered using some form of clustering approaches (e.g. K-mean) based on key characterisetic for clustering the sites.

Section 3 Method - It seems that the model interconnect different site only as part of step 1 (phase randomisation/perturbation applied throught the medium of white noise). All the remaing steps (1-4 steps) are applied independently across all the sites. I think the approach is appropriate. A separate Kappa distribution is fitted for each day for a 30-day window to factor in seasonal differences. I have a minor concern here, What is the motivation for the selection of a 30 day window, how does it effect data with different seasonal periods across different sites (e.g. may be one site having monthly seasonality but other having weekly seasonal characteristics or say over a three months period).

Section 4 A robust evaluation has been conducted that includes careful seletion of sites (distinct and representative). Statistics used for comparision are appropriate and results are well explained. Some minor comments - Figure 9 – Visually observed and simulated looks in good agreement for occurance of POT events but for a robust comparision some measurements should bave been used in parallel. Figure 10 gives a good illustration of how spatial dependencies could be effected among th sites with respect ot the Euclidean distance. However, for the readers benefit it would be appreciate if Authors' consider to provide few sentences to explain F-madograms plots, specifically, what should a relative difference of 0.05 in observed and simulated values should be interpreted.

Section 5 and 6 – Overall good and capture key aspects of the paper.

Reference [1] Manuela I. Brunner, András Bárdossy, and Reinhard Furrer, Technical note: Stochastic simulation of streamflow time series using phase randomization, Hydrol. Earth Syst. Sci., 23, 3175–3187, 2019.

Please also note the supplement to this comment:

https://www.hydrol-earth-syst-sci-discuss.net/hess-2019-658/hess-2019-658-RC2-supplement.pdf

---

## Author Comment (AC1) · 17 Mar 2020

**Reviewer 1**

GENERAL COMMENTS
The manuscript proposes a wavelet-based phase randomization approach to generate multisite hydrological extremes, which is argued to be able to capture both spatial dependencies and non-stationarities. The stochastic simulated data is reconstructed with the same values as original data in another temporal order, and this assures the reproduction of temporal dependence, extremes and nonstationarities.
In addition, the multiplication of the same set of random phases to all the investigated sites ensures spatial dependences. The paper is clearly written, and the idea is well explained. However, I have listed some comments below which needs to be addressed before the manuscript is considered further for publications.

**Reply:** *Thank you very much for your comments, which we address point-by-point below.*

**Major comments:**
COMMENT1: The spatial dependence can be simulated by using the same randomized phases for multiple time series among sites. You claimed this in your introduction and Section 3.2 step 1 without detailed explanation and references. From my understanding, the phases represent the time of the timing of changes or variations (in your case, it is extreme events), so if you use the same randomized phases, it is expected that there will be spatial dependence of extremes among the sites you investigated. Thus, could you give explanations or illustrations on this statement.
**Reply:** *Prichard and Theiler (1994) and Schreiber and Schmitz (2000) describe that the phases of all sites have to be randomized in the same way to preserve cross-correlations for multivariate time series. We added the two references to the description of Step 1. Chavez and Cazelles (2019) later proposed a surrogate approach for multivariate time series for a dataset of weekly measles notifications and an electroencephalographic recording where they apply a phase randomization procedure using random phases extracted from a random, normally distributed time series. We also added this reference to the corresponding section.*
*Figure 1 in this response to the reviewer shows on the example of the four stations in the Pacific Northwest (see Figure 1 in paper) that the observed cross-correlation would not be captured by the simulations if the phases were randomized for each catchment individually instead of using the same set of randomized phases across all catchments.*
**Modification: p.4, l.97; p.6, l.151**

[Figure]

*Figure 1: Comparison of observed (black) and simulated (orange) cross-correlation functions (ccfs) for the daily discharge values of the four catchments (i--iv) in the Pacific Northwest. 20 simulations were generated for each site individually, neglecting spatial dependence.*

COMMENT2: Section 3.2, step 1 shows how the random phases are computed from a random discharge time series from a normal distribution with mean 0 and standard deviation 1. Could you explain the specific reason to choose normal distribution or have you tested with other distribution for example, gamma and kappa distribution?

**Reply:** *It is correct that we compute the random phases from a normal distribution. We have also tested the kappa distribution, which did not change the results. We decided to go with the normal distribution as such a distribution was also used by Chavez and Cazelles (2019) who proposed a surrogate approach for multivariate time series for a dataset of weekly measles notifications and an electroencephalographic recording. This reference was added to the description of Step 1.*

**Modification: p.6, l.151**

Another similar question is about the selection of wavelet family and scale. In this study, you use Morlet wavelet, what is the specific reason to use this wavelet family, and how about other wavelet families, e.g., Paul, DOG and Marr (Torrence and Compo, 1998)? The sensitivity of your approach to the selection of wavelet family and scale? For the application of wavelet method in real world, the selection of wavelet family and scales is of great importance.

**Reply:** *Because of the phase randomization step, our application requires a complex wavelet with an imaginary part in addition to a real part. The DOG wavelet family is real valued while the Morlet and Paul wavelets are complex valued [Torrence and Compo, 1998]. Among the complex valued wavelets, we use the Morlet wavelet because it has been found suitable for hydrological applications in previous studies [Lafrenière and Sharp, 2003; Labat et al., 2005; Schaefli et al., 2007]. It provides a better frequency localization than other complex wavelets such as the Paul wavelet [Torrence and Compo, 1998]. We added the additional references (Lafreniere et al. 2003 and Labat et al. 2005) to the text. It is important that the number of scales is chosen high enough to allow for a fine resolution [Torrence and Compo, 1998]. Using e.g. only 20 scale results in reconstructed time series that do not reflect all the necessary detail. We chose 100 scales because further increasing the number of scales no longer improves reconstruction performance.*

**Modification: p.4, l.120-121**

Additionally, I am unable to visualise where exactly the random phases go back into equation 3?
**Reply:** *The complex part of the wavelet transform Wn(h) comes in via the complex conjugate \* in Equation 2. During back-transformation in Equation 3, it comes back in when deriving the real part at each scale h through R(Wn(hj)).*

What should have been the values of these phases with the observed data in the first place?
**Reply:** *The phases are uniformly distributed over the range of -π to π, which was specified in the text.*
**Modification: p.7, caption of Figure 2**

Why should these phases be the same? I suspect the only impact these phases have is on spatial dependence. In that case why are the majority of results that are presented focusing on temporal dependence attributes? Figure 6 gives some flavour of the multi-suite stochastic generation. It looks troublingly similar to the observations. What would happen if non-identical phases used across all the sites? These questions are important to address to establish the contribution here is an improved representation of spatial dependence compared to other alternatives.
**Reply:** *The random phases have to be the same across all stations to retain spatial dependence, which is mentioned in the text and is why the spatial aspect receives a lot of attention in the validation part of the manuscript (Figures 6-11, i.e. half of the figures). Figure 6 gives an impression of what the simulated time series look like for three regions with four stations each. While the general distributional and temporal characteristics of the series are retained by the simulations, the simulations do not reproduce observed events but instead generate new series of potential successions of events. Figure 7 shows the cross-correlation functions among pairs of stations for the four catchments in the Pacific Northwest. If non-identical phases were used, these spatial correlations would not be reproduced as demonstrated in Figure 1 shown above. Figure 8 summarizes spatial dependencies over the whole distribution for all pairs of stations (671 times 671) in a variogram. Figure 9 gives an overview of spatio-temporal characteristics of simulated flood events and Figure 10 summarizes spatial dependencies for all pairs of stations (671 times 671) via the F-madogram, a measure of extremal dependence. Also the last Figure (11) focuses on the spatial dependence aspect by comparing observed with simulated tail dependence coefficients, which are again computed for all pairs of stations. We clarify the description of the evaluation procedure by specifying that the general spatial dependence structure and the spatial dependence in high extremes is evaluated. Figure 1 in this response to the reviewer shows that spatial correlations are not retained if non-identical phases are used across all sites.*
**Modification: p.9, l.199-200**

Regarding other alternatives, is there a possibility of comparing this method with the commonly used spatial stochastic generators such as SPIGOT or other equally worthy alternatives? I would like to see the variogram figure possibly expanded to also show alternate outcomes using other formulations of spatial dependence (i.e. different ways of specifying phases).
**Reply:** *We rerun the simulations for the whole set of stations without using the same phases across all stations but generating a set of random phases for each station individually. Figure 2 in this response to the reviewer shows that spatial dependence is modeled much worse than when randomizing the phases for all catchments in the same way (Figure 8 in manuscript).*
*We agree that a comparison of different spatial stochastic generators with respect to how they represent spatial extremes would be interesting and valuable. In order to make such a model comparison study beneficial for the community, ideally a broad range of models ranging from continuous to event-based models should be compared as there exists no commonly accepted benchmark/reference model. Such a comparison goes beyond the scope of this manuscript and should be addressed in a separate study.*

[Figure]

*Figure 2: Comparison of observed (black) and simulated (orange) variograms for 20 simulation runs where the phases were randomized for each station individually.*

COMMENT3: Section 3.2, step 4 mentions how the reconstruction is done by using the inverse wavelet transform (Eq. (3)) combining the derived random phases and the amplitudes in previous steps. Could you write the combination explicitly in the from of the equation (i.e., include the random phases in the Eq. (3))?

**Reply:** *The complex part of the wavelet transform Wn(h) comes in via the complex conjugate \* in Equation 2. The complex conjugate is derived using the modulus and argument, i.e. the phases, of the complex numbers contained in the Morlet wavelet (Equation 1). The wavelet coefficients Wn(h) resulting from the transform are also complex numbers where the argument of the complex wavelet value Wn(h) corresponds to the wavelet phase. During back-transformation in Equation 3, the phases come back in when deriving the real part at each scale h through R(Wn(hj)).*

Minor comments: Line 147: h=100 wavelet scales, I think you mean number of wavelet scales is 100, not the scale itself equals 100. Wavelet scales should be hj = h0*2^(j*dj), j=0,1,: : :J. Or is this some parameter that is specific to a continuous wavelet transform? Why is it of relevance, what impact does it make, and why use the same value for all locations is something that should be discussed.

TORRENCE, C. & COMPO, G. P. 1998. A Practical Guide to Wavelet Analysis. Bulletin of the American Meteorological Society, 79, 61-78.

**Reply:** *Thank you for pointing this out, yes, we intended to refer to the number of wavelet scales. This was adjusted in the text. The number of scales can be freely chosen in the case of the continuous wavelet transform. A high number of scales compared to a low number of scales allows for a finer resolution* [*Torrence and Compo*, 1998] *and for a detailed reconstruction of the original time series.*
**Modification: p.8, l.173**

**References used in this response to the reviewer**

Chavez, M., and B. Cazelles (2019), Detecting dynamic spatial correlation patterns with generalized wavelet coherence and non-stationary surrogate data, *Sci. Rep.*, *9*(1), 1–9, doi:10.1038/s41598-019-43571-2.

Labat, D., J. Ronchail, and J. L. Guyot (2005), Recent advances in wavelet analyses: Part 2 - Amazon, Parana, Orinoco and Congo discharges time scale variability, *J. Hydrol.*, *314*(1–4), 289–311, doi:10.1016/j.jhydrol.2005.04.004.

Lafrenière, M., and M. Sharp (2003), Wavelet analysis of inter-annual variability in the runoff regimes of glacial and nival stream catchments, Bow Lake, Alberta, *Hydrol. Process.*, *17*(6), 1093–1118,

doi:10.1002/hyp.1187.

Prichard, D., and J. Theiler (1994), Generating surrogate data for time series with several simultaneously measured variables, *Phys. Rev. Lett.*, *73*(7), 951–954.

Schaefli, B., D. Maraun, and M. Holschneider (2007), What drives high flow events in the Swiss Alps? Recent developments in wavelet spectral analysis and their application to hydrology, *Adv. Water Resour.*, *30*(12), 2511–2525, doi:10.1016/j.advwatres.2007.06.004.

Schreiber, T., and A. Schmitz (2000), Surrogate time series, *Phys. D Nonlinear Phenom.*, *142*(3–4), 346–382, doi:10.1016/S0167-2789(00)00043-9.

Torrence, C., and G. P. Compo (1998), A practical guide to wavelet analysis, *Bull. Am. Meteorol. Soc.*, *79*(1), 61–78.

---

## Author Comment (AC2) · 17 Mar 2020

**Reviewer 2: Dr. Sandhya Patidar**

The paper presents a continuous wavelet-based phase randomisation approach for the stochastic generation of streamflow time series. This is an interesting paper that substantially extends the original techniques developed in [1] for stochastic simulation of streamflow using Fourier transformation based phase randomisation. The original method, presented in [1] is associated with certain limitations such as application of Fourier transformation does not account for the non-stationarities in time series and resulted in an underestimation of spatial dependencies (e.g. cross-correlation) in both daily discharge and extreme events across the multiple sites. The present paper address some of the limitations observed in [1] by replacing Fourier transformation with a complex wavelet-based approach.

The efficiency of original method has been evaluated to generate realistic series for distributional and temporal correlation characteristics and is validate through the application across four catchments in Switzerland. The proposed model is applied to a large dataset of 671 catchments in the contiguous United States and the efficiency of the model has been evaluated by assessing its ability to capture distributional and temporal characteristics at individual sites along with the spatial dependencies across the multiple sites, and extreme events (floods) including duration and volume.

**Some general comments**

Section 1 - Considering the theme of paper that signifies the application of wavelet approach, the Introduction section presents an interesting critical review on recently developed/applied modelling schematics that involves wavelet-based phase randomization as a key approach.

Line 70-73: To add clarity it would be helpful if the authors' team add some brief explanation on how data normalisation procedure and back transformation impacts the spatial dependencies.

**Reply:** *Thank you for pointing out the need for clarification. We specify that 'This weakening is because phase randomization preserves the cross-correlation in the normal domain but not necessarily in the domain of the original distribution as linear correlation is not invariant under non-linear strictly increasing transformations.'*

**Modification: p.3, l.71-73**

Line 87-88: Please add some clarity on how continuous wavelet transform is more effective than a discrete wavelet transforms in minimising/overcoming issues around the long-term periodicities and/or non-stationarities. Is there any specific studies carried out to investigate such issues.

**Reply:** *The discrete wavelet transform only allows for real wavelet functions while the continuous transform allows for complex wavelet functions. Stochastic approaches randomizing only real-valued amplitudes have been shown to have problems with the reproduction of non-stationarities [Breakspear et al., 2003]. Chavez and Cazelles (2019) show that the randomization of phases (information stored in the complex-valued coefficients) allows for the generation of non-stationary time series. We rephrased the whole paragraph to clarify that the advantage of using a continuous instead of a discrete transform comes from the additional phase information gained when using complex wavelet functions which are only available for the continuous transform.*

**Modification: p.3, l.83-83 and l.89-93**

Section 2 - Theoretical background section provides suffiecient details on the wavelet decomposition approach.

**Reply:** *We are glad that you consider the background section to provide sufficient detail.*

Section 3 Data – For illustration and validation purposes, dataset are organised in three different region based on the general hydrological characteristics. It is not clear which specific properties has been used and how rigorously they have been appied. I think, this work could have benefitted if Authors' have considered using some form of clustering approaches (e.g. K-mean) based on key characterisetic for clustering the sites.

**Reply:** *Thank you very much for this suggestion. We indeed applied a clustering procedure to define these regions which are similar in terms of their flood characteristics. The clustering was applied on a distance matrix computed from the F-madogram, which is a measure of extremal dependence, between pairs of stations. The clustering was applied to the 671 catchments and resulted in 15 clusters among which we selected 3 for illustration purposes. Our manuscript, where we describe this clustering procedure and the resulting clusters, is currently under review and we will include its reference here if it receives a DOI before this manuscript eventually goes into press.*

Section 3 Method - It seems that the model interconnect different site only as part of step 1 (phase randomisation/perturbation applied throught the medium of white noise). All the remaining steps (1-4 steps) are applied independently across all the sites. I think the approach is appropriate. A separate Kappa distribution is fitted for each day for a 30-day window to factor in seasonal differences. I have a minor concern here, What is the motivation for the selection of a 30 day window, how does it effect data with different seasonal periods across different sites (e.g. may be one site having monthly seasonality but other having weekly seasonal characteristics or say over a three months period).
**Reply:** *Thank you for acknowledging the appropriateness of our approach. We chose a window of 30 days to temporally pool data prior to the estimation of the daily parameters of the kappa distribution. The idea of the smoothing is to reduce the effect of sampling uncertainty and to reduce day-to-day variability in parameters under the assumption that the distribution of flow on day x is unlikely to be substantially different from the one on day x+1. A value of 30 days was chosen for the moving window to enable sufficient smoothing in the parameter space and to ensure a sufficient sample size for the estimation of the four parameters. The moving-window nature of the approach allows for capturing seasonal and partly also weekly variations. We clarified in the method description that this approach corresponds to a moving window approach.*
**Modification: p.8, l.165**

Section 4
A robust evaluation has been conducted that includes careful seletion of sites (distinct and representative). Statistics used for comparision are appropriate and results are well explained.
**Reply:** *Thank you for appreciating the representativeness of our evaluation.*

**Some minor comments**
Figure 9 – Visually observed and simulated looks in good agreement for occurance of POT events but for a robust comparision some measurements should bave been used in parallel.
**Reply:** *We computed differences in the mean inter-event duration (i.e. time elapsing between two successive events) of observed and simulated series and the standard deviation of inter-event duration for events where 1, 2, and 3 stations were jointly affected, respectively. We find that over all three regions, relative differences in mean and standard deviation of inter-event duration lie mostly below 10%. However, we find that a visual comparison is most effective here to demonstrate the value of the simulation approach.*

Figure 10 gives a good illustration of how spatial dependencies could be effected among th sites with respect ot the Euclidean distance. However, for the readers benefit it would be appreciate if Authors' consider to provide few sentences to explain F-madograms plots, specifically, what should a relative difference of 0.05 in observed and simulated values should be interpreted.
**Reply:** *We specified in the figure caption that the F-madogram is a measure of extremal dependence. We also state that an overestimation of spatial dependence means that a pair of stations co-experiences more joint floods than in the observations. What a difference of 0.05 means in terms of differences in regional hazard estimates is hard to say and would need to be investigated in a proper study where the stochastic simulations are used to estimate regional flood hazard.*
**Modification: p.17, caption of Figure 10**

Section 5 and 6 – Overall good and capture key aspects of the paper.
**Reply:** *Thank you.*

Reference
[1] Manuela I. Brunner, András Bárdossy, and Reinhard Furrer, Technical note: Stochastic simulation of streamflow time series using phase randomization, Hydrol. Earth Syst. Sci., 23, 3175–3187, 2019.

**References used in this response to the reviewer**

Breakspear, M., M. Brammer, and P. A. Robinson (2003), Construction of multivariate surrogate sets from nonlinear data using the wavelet transform, *Phys. D Nonlinear Phenom.*, *182*(1–2), 1–22, doi:10.1016/S0167-2789(03)00136-2.

Chavez, M., and B. Cazelles (2019), Detecting dynamic spatial correlation patterns with generalized wavelet coherence and non-stationary surrogate data, *Sci. Rep.*, *9*(1), 1–9, doi:10.1038/s41598-019-43571-2.

---

## Author Comment (AC3) · 17 Mar 2020

**Commentator Tsoukalas**

**Introduction**

This is an interesting work aiming to provide a method (as well as implement it in an R package; function called PRSim.wave within PRsim R package) for the simulation of multivariate hydrological processes (for now, focusing on streamflow) - which according to the results presented by the Authors has a good potential, requiring yet some improvements.

In general, I find the manuscript well-organized and straightforward to understand, yet in my view, there are several points that require the Authors attention.

All comments and suggestions are meant to be constructive and aim to improve the quality of the manuscript, as well as the findings obtained.

**Reply:** *Thank you for your detailed comments, which we address below.*

**Comments**

- **L6-7.** I suggest to write: "To do so, we propose the stochastic simulation approach called Phase Randomization Simulation using wavelets (here after called *PRSim.wave*) which combines… ".
  **Reply:** *We added brackets around PRSim.wave.*
  **Modification: p.1, l.7**

- **L11.** To avoid confusing the reader, and provided that a few lines above it is mentioned that "We apply and evaluate *PRSim.wave* on a large set of 671 catchments in the contiguous United States.", I suggest to write: "…at multiple sites (up to four)…"
  **Reply:** *We mean to say that we evaluate the set on 671 catchments. However, it is correct that we focus on three sets of four stations each in order to present a few more detailed results. Since we show that the approach works for very large datasets, we think that a generalization to 'multiple sites' is justified.*

- **L34.** I wonder what are the "potential non-stationarities" mentioned by the Authors? Are you refereeing to the typical cyclostationary behavior exhibited by hydrological processes? Given the opportunity, and as a side note, I would like highlight that stationarity is an essential tool for inferencing from data (e.g., model fitting). Stationarity should not be seen as a shortcoming, nor dead. Non-stationarity implies non-ergodicity, which in turn makes inference from observed data impossible, unless of course the deterministic dynamics of the process are known; which in my understanding, is never the case in hydrological sciences. On this topic, I recommend the recent work of Serinaldi et al. (2018), with emphasis on section 4.2, as well as the works of Koutsoyiannis and Montanari, (2007), (2015), Lins and Cohn (2011), Matalas (Matalas, 2012), and Montanari and Koutsoyiannis (2014), that argue in favor of stationarity. See also the very interesting, note of Harry F. Lins. As Harry F. Lins concludes his note:
  Stationarity ≠ static
  Non-stationarity ≠ change (or trend)
  **Reply:** *What we state in this manuscript is that PRSim.wave is able to reproduce non-stationarities in the data. This is shown in Figures 4e. It is shown there that the wavelet power corresponding to different scales varies over time, i.e. the importance of different cyclical features varies over time. We are neither saying that stationarity is dead nor that it is a shortcoming. In contrast, we point out that developing stochastic models that can handle non-stationarities is important.*

  **L36.** I am not sure what is the meaning of "continuous" here? Can you please elaborate/specify? Also, some references would be useful.

**Reply:** *We specify that continuous approaches correspond to discrete-time models in the stochastic literature.*
**Modification: p.2, l.36-37**

- **40.** Although I understand its rationale, I am not a big fan of the now-typical classification of stochastic models on "parametric and non-parametric", since in my view, there is no model without parameters. Typically, the literature uses the term "non-parametric" to refer to approaches that use some kind of resampling mechanism (e.g., k-nn algorithm) and/or "non-parametric" distribution functions (e.g., kernel-based approximation of the density function) to generate synthetic data. But, one should take a moment and think, are these really non-parametric? Isn't the $k$ (i.e., the number of nearest neighbors) in k-nn algorithm a parameter? Isn't the choice of the kernel smoothing function (e.g., normal, epanechnikov, box, triangle) a parameter? Isn't the bandwidth of the kernel a parameter (also called the smoothing parameter)? Aren't the data *per se* used as parameters (e.g., when a non-parametric method relies on the sampling from the empirical CDF or kernel-based CDF. What if we have new data or alter a few? Does the model change?)? Having said these, I suggest to the Authors to reconsider using the employed classification, as well as review the recent literature (e.g., Serinaldi and Kilsby, 2014; Tsoukalas et al., 2019) for finding an alternative classification.
  **Reply:** *We here understand a parametric approach as an approach where a model is fitted to the data. We agree, however, that the choice of k in a nearest-neighbor algorithm might also be viewed as a parameter in a wider sense. We would, however, not go as far as to say that the data are parameters. However, they are subject to sampling uncertainty, which influences model fitting.*

- **L41-43.** The Authors write: "…and temporal disaggregation models such as fractional Gaussian noise models (Mandelbrot, 1965), fast fractional Gaussian noise models (Mandelbrot, 1971), broken line models (Mejia et al., 1972), and fractional autoregressive integrated moving average models (Hosking, 1984).".
  To clarify, these are not disaggregation models, but models able to simulate processes exhibiting long-range dependence (particularly, designed to simulate fractional Gaussian noise (fGn) processes or else, processes exhibiting Hurst behavior). See a similar discussion in the introduction section of Tsoukalas et al. (2018b).
  **Reply:** *We rephrased the sentence as follows: 'Parametric models include autoregressive moving average (ARMA) models and their modifications (Stedinger1982, Paplexiou2018), and fractional Gaussian noise models (Mandelbrot1965) comprising fast fractional Gaussian noise models (Mandelbrot1971), broken line models (Mejia1972), and fractional autoregressive integrated moving average models (Hosking1984).'*
  **Modification: p.2, l.42**

- **L44-45.** The Authors write: "Nonparametric models are based on disaggregation and resample from the data with perturbations and include…".
  I think that this statement can be confusing, and needs some refinement. "Non-parametric" models are not necessarily based on the notion of disaggregation. Of course, the literature offers "non-parametric" disaggregation methods (e.g., Lee et al., 2010; Tarboton et al., 1998), but this does not makes all "non-parametric" methods, methods that "based on disaggregation and resample". Further details on disaggregation methods can be found on the seminal works of Valencia and Schakke (1973), and Mejia and Rousselle (1976), as well in the work of Koutsoyiannis (2001) who provide a detailed overview on the subject. For a more recent overview and discussion on the topic of disaggregation and multi-temporal simulation see also work of Tsoukalas et al. (2019).
  **Reply:** *Thank you for pointing out that this statement is confusing. We rephrased the*

*sentence to 'Nonparametric models include kernel density estimation (Lall1996, Sharma1997) and various bootstrap approaches…'*
**Modification: p.2, l.44-45**

- **L48-49.** The Authors write: "…but none of these time domain methods can capture the spectral properties of the observed time series (Erkyihun et al., 2017).".
  In my view this statement is a bit confusing, requiring the Authors attention, for two reasons.
  1) A timeseries (i.e., a sequence of observations ordered in time) does not has spectral properties, it exhibits some form of dependence structure (which can be quantified using statistics/stochastics, e.g., through the empirical correlation coefficients and the empirical spectrum). What has spectral properties is the stochastic process that it is assumed that generated the observed timeseries.
  2) Having said the above, and since correlation and spectrum are interrelated quantities, if a model is capable of reproducing the process's correlation structure it also reproduces its spectrum (and vice versa). For further details and references, see my previous comment (Tsoukalas, 2019) on a recent work co-authored by the first Author of this work.
  **Reply:** *Thank you for pointing out that this statement was confusing. We removed the first part of this sentence and now state 'and the representation of spatial dependence in such time-domain models is challenging.'*
  **Modification: p.2, l.48**

- **L49-50.** In my view the sentence "Furthermore, these time-domain models struggle with the representation of spatial dependence" is a bit "strict", since as far as I see it, there is no struggle, but many research efforts (past, and new).
  The stochastic hydrology literature offers several "time-domain" models that can simulate parsimoniously multivariate processes, including both stationary and cyclostastionary processes (e.g., Efstratiadis et al., 2014; Koutsoyiannis, 2001, 2000), reproducing also the moments of the observed processes (typically up to third order). Further to these models/methods, more recent approaches allows the parsimonious simulation of multivariate stationary and cyclostationary processes with any marginal distribution and correlation structure (Kossieris et al., 2019; Tsoukalas, 2018; Tsoukalas et al., 2018a, 2018b), also in a multi-scale context (Tsoukalas et al., 2019). Apart from the last work, for another multi-scale and multivariate simulation study involving daily rainfall at 4 sites the Authors are referred to Appendix D, section D.2, of Tsoukalas (2018). Therefore, taking into consideration the above-mentioned works I would suggest the Authors to revise the sentence accordingly, as well as provide some references.
  **Reply:** *We rephrased the sentence to 'and the representation of spatial dependence in such time-domain models is challenging.'*
  **Modification: p.2, l.48**

- **L54-55.** The Authors write: "In contrast to time-domain models, frequency-domain models allow for the simulation of surrogate data with the same Fourier spectra as the raw data".
  This can be also true for time-domain methods (but not a good modelling practice in either of the two cases; see below). For instance, if one employs an AR or MA model of high order can simulate a realization of a process exhibiting exactly the empirical autocorrelation coefficients up to the order dictated by the model. However, this is not a good modeling practice since it is well-known that the empirical estimators of auto- (and cross-) correlation coefficients are (downward) biased (Beran, 1994; Koutsoyiannis, 2003, 2000), especially in the case of long-range dependence, short samples, and large lags. See also Matalas (1967 p. 945) who remark that:
  "*Parameters that are determined in terms of high order moments of large time lags are subject to large standard errors and consequently large operational biases. Operational*

*biases can never be eliminated, but they can be minimized by the use of regionalization to account for the temporal and spatial variations inherent in the historic sequences…”.*

Of course, the same applies for the empirical estimators of spectrum (see the comparative work of (Dimitriadis and Koutsoyiannis (2015)). Note that this kind of approaches are not parsimonious (since all the empirical estimates used in model fitting are essentially model parameters). To cope with these, the recent literature (Kossieris et al., 2019; Tsoukalas et al., 2019, 2018b), as well some works already cited in the manuscript (i.e., Papalexiou (2018)), has leaned towards the use of parametric models (e.g., with two or three parameters) to parsimoniously describe the dependence structure of the processes. The Authors are referred to the work of Koutsoyiannis (2000) which in my view popularized that idea in hydrological domain, also introducing a parsimonious two-parameter auto-correlation structure. It is also interesting to note the work of Papalexiou (2018) (already cited in the manuscript), who employed the functional form provided by the survival function of a distribution to define several auto-correlation structures.

**Reply:** *We weakened the statement and rephrased the sentence to: 'In contrast to most time-domain models,..'*

**Modification: p.2, l.53**

- **L70-72.** The Authors write: "In addition, it may help to improve the representation of spatial dependencies because it does not require a transformation to the normal distribution and back to the original, skewed distribution, which usually weakens spatial correlations (Embrechts et al., 2010)."

  First, the comment on "weaken spatial correlations" applies for all "types" of correlations (that emerge from the mapping/transformation from the Gaussian to the actual domain) – not only spatial. Particularly, in the case of stochastic processes, it also applies for the auto-correlation structure of a stationary processes, as well as for the season-to-season correlations of a cyclostationary process (Tsoukalas et al., 2018a, 2017). It also holds for multivariate cases. However, I am afraid that I cannot see the improvement of the representation of spatial dependencies mentioned above by the Authors. The cross-correlations as well as the auto-correlation are still not accurately reproduced (see my comments below on the results/plots). It is my understanding that a previous comment of mine (Tsoukalas, 2019) on a recent work co-authored by the first Author of this work holds also for this method. This is due to the following:

  **Reply:** *We here specifically talk about the effect of back transformation on spatial correlation because this is the focus of this study. As mentioned in the manuscript the wavelet transform employed here does, compared to the Fourier transform, not necessitate a transformation to the normal domain, and a back transformation to the domain of the skewed distribution, which improves the representation of spatial correlations as compared to the Fourier transform based approach [Brunner et al., 2019]. Figure 1 shown in the response to reviewer 1 illustrates that spatial correlations are not well represented if phases are randomized for each station individually. In contrast, the use of the same random phases for all the stations as proposed in this manuscript leads to a nice reproduction of spatial dependencies as illustrated e.g. by the cross-correlation functions shown in Figure 7 in the manuscript.*

- **L145.** The Authors write: "Derivation of random phases: A random discharge time series (white noise) of the same length as the input series is sampled from a normal distribution with mean 0 and standard deviation 1."

- **L170-173.** The Authors write: "Transformation to kappa distribution: The simulated values are transformed to the kappa domain using the fitted daily kappa distributions from Step 2. For each day, a random sample is generated from the fitted, daily kappa distribution. The simulated values are replaced by the values generated from the kappa distribution using rank-ordering. This procedure is repeated for each day in the year."

Based on the above the method presented herein depends on an auxiliary Gaussian process and uses the target ICDFs, as well as the rank-correlations to establish the (auto- and cross-) dependence structure. It is reminded that such a procedure will preserve the ranks correlation coefficients (which do not depend on the marginals) but not the Pearson's, (which depends on the marginals; since it involves the cross-product moment of the among the variables). For further details the Authors are referred to the comment mentioned above, as well as in the references therein. It is my understanding that the mechanics of the method that dictate the preservation of ranks is the reason why the auto- (and cross-) correlations are not so well reproduced by the proposed method.

**Reply:** *The results shown throughout Figures 7-11 demonstrate that the wavelet-based phase-randomization approach we present in this manuscript is well capable of reproducing observed spatial dependencies which is not the case when randomizing the phases of the Fourier transform as shown in Brunner et al.* (2019)*.*

- **L149 (and elsewhere)**: The use of Kappa distribution. As mentioned in a previous comment of mine in HESS related with a work co-authored by an Author of this manuscript there are few complications worth considering when using the Kappa distribution. The following comments are excerpted with minimum or no modifications at all from Tsoukalas (Tsoukalas, 2019).

  1. Since you are using the Kappa distribution it could be insightful to mention that under certain parameter combinations, this distribution may lead to infinite moments. This can be a delicate issue, since if the fitted distribution exhibits infinite variance then the Pearson's correlation cannot be defined (the denominator contains the variance), and thus the proposed model (as well as many other models) cannot be used. This situation is discussed in section 3.4 of Tsoukalas et al. (2018b; and references therein), where it is advocated (based on empirical, as well as theoretical reasoning) that physical processes are characterized by finite variance (Koutsoyiannis, 2016). Particularly, if $X$ is a Kappa-distributed random variable, and $\mu_r = [X_r]$ denotes the $r_{th}$ raw moment, as discussed in Hosking (1994), and elsewhere, the existence of the $r_{th}$ depends on the values of $h$ and $k$. Specifically, the moments exist: for all $r$ if $h≥0$ and $k≥0$ for $r<−1hk/$ if $h<0$ and $k≥0$,and for $r<−1k/$if $k<0$

  It is also interesting to mention that Hosking (1994) notes that the first four moments cannot uniquely determine the parameters of the distribution, since some combinations of moments (expressed by skewness and kurtosis coefficients) correspond to different pairs of $h$ and $k$.

  2. How do you handle negative values? As far as I am aware the left (and right) support of Kappa distribution is not necessarily zero (e.g., when $k=0$ and $h≤0$, then the supports of the distribution are, $−∞<x<∞$; see Hosking (1994)). In any case, the generation of negative values can be eliminated by using a distribution function defined in the positive real line. Particularly, I would suggest the investigation/use of the Generalized Gamma and Burr type-XII distributions, which are more parsimonious (they entail three parameters; instead of four as in Kappa) and were found adequate for modelling of hydrometeorological variables; particularly rainfall (e.g., Papalexiou and Koutsoyiannis, 2016). Examples of their use within the context of stochastic modelling can be found the work Papalexiou (2018), as well as in Tsoukalas et al. (2019, 2018b) and Tsoukalas (2018).

  **Reply:** *We tested the Burr type XII (validation results see Figure 1 in https://www.hydrol-earth-syst-sci-discuss.net/hess-2019-142/hess-2019-142-AC1-supplement.pdf) and generalized Gamma distributions, which were, however, not flexible enough to model the marginal distributions of the daily discharge values and led to unrealistically extreme high flows in the simulations and will therefore not be considered as alternatives to the kappa distribution. The R-package PRSim, however, allows for the implementation of any distribution chosen by the user as specified in the section code and data availability. We specify that negative simulated values are replaced by 0, which corresponds to the use of a*

*bounded kappa distribution.*
**Modification: p.8, l.178**

- **L158-160.** The Authors write: "We fit a separate distribution for each day to take into account seasonal differences in the distribution of daily streamflow values. To do so, we use the daily values in a 30-day window around the day of interest."
  Can you please elaborate on this? Just to be sure, for each "site" and for each day of the year a Kappa distribution has been fitted with different parameters? If this is the case, just for the marginal behavior, and for each site you fitted Kappa 365 times, which implies that the model has 365×4 (the number of parameter of Kappa)=1460× the number of sites, parameters (not accounting those for the specification of the auto and cross-dependence structure of the process). If this is the case, I am afraid that this is not a parsimonious model, something that should be clearly stated in the manuscript (also mentioning the total number of its parameters).
  Also, I don't think that it is reasonable to assume that days belonging in the same month (e.g., the 19th and the 20th of August) have different marginal distribution (although, I have seen stochastic simulation related works following that approach, I am not aware of any paper supporting this assumption). The standard approach for daily (or finer time) scales is to consider stationarity within the monthly interval (i.e., in the case of daily data consider that all days belonging in the month have the same marginal distribution). An arguably more parsimonious approach, since in this case the total number of parameters for the marginal behavior would be 12×4 (the number of parameter of Kappa)=48× the number of sites. Note that the number of parameters could be further reduced by using, instead of Kappa, alternative 2- or 3-parameter distribution models.
  **Reply:** *Yes, the kappa distribution was fitted for each day of the year. A moving window approach was used to reduce the effect of sampling uncertainty and to reduce day-to-day variability in parameters under the assumption that the distribution of flow on day x is unlikely to be substantially different from the one on day x+1. We acknowledge in the discussion section that ' it [the model] requires the fitting of many parameters, which make the model non-parsimonious (Koutsoyiannis2016).*

- **L162-164.** The Authors write: "In a few regions with many zero discharge values (e.g. some catchments in the Great Plains) fitting the kappa distribution is not possible and we therefore use the empirical distribution instead."
  This is a work-around that could work, but I wonder, why not use an alternative distribution model (e.g., zero-inflated or mixed) that can model simultaneously both the discrete (i.e., probability of no discharge) and continuous part (i.e., distribution of non-zero discharge) of the process? Also, can you provide an estimate on the number of cases where the empirical distribution is employed instead of Kappa?
  **Reply:** *This work around was necessary for those stations in the Great Plains showing an intermittent flow regime with many 0 values. If desired, the R-package PRSim allows for the implementation of an alternative distribution as specified in the data and code availability section.*

- **L218-219.** The Authors write: "the seasonal (3d) and monthly distributions (3e–g) are well captured by the simulations."
  This is a confusing description of the plots. The (3e–g) plots show that some seasonal summary statistics are reproduced (i.e., monthly mean, monthly maxima, monthly minima), not the seasonal distributions (to do so you need alternative plots, comparing the empirical distribution of each month with the corresponding theoretical one). Also, I don't understand what plot 3d shows? What does "seasonal statistics" means in the title of this plot? What does the phrase "the seasonal discharge distribution" means at the legend of Figure 3? These

are box-plots, and by no means should be used to compare distribution functions (they provide way too few information - about specific quantiles).

**Reply:** *We rephrased this sentence to 'the seasonal and monthly distribution characteristics'. The boxplots are used here to give an impression of the good match between median, upper, and lower quartile. We see the information provided by the boxplots as a proxy for important distribution characteristics. Tails of the observed and simulated distributions are not expected to perfectly match as the theoretical distribution allows for the extrapolation to unobserved values.*

**Modification: p.9, l.225**

- **L220-222.** The Authors write: "the temporal correlation characteristics (4a–c), …. are well captured by the simulations as well."
  In my view there is an important difference between the simulated and empirical autocorrelation coefficients. For instance, by eye-balling the median of the orange lines (simulated) for time lag 10 we get a value about equal to 0.5, while the observed one is 0.4. This should be clearly stated in the manuscript. For the readers convenience, I also suggest the inclusion of a line depicting the median of the simulated quantities (orange lines). Providing lines for a low and high quantile (say 0.05 and 0.95) would be nice also. This comment applies for all similar plots throughout this manuscript.
  **Reply:** *As illustrated by Figure 4, the simulated temporal correlation characteristics are very close to the observed ones. We depict individual lines for individual realizations to communicate the variability inherent in the simulations to the reader.*

- **L223-225.** The Authors write: "Both high- and low- extremes are realistically modeled as illustrated by the distributions of the above and below threshold events of the four catchments in the Pacific Northwest (Fig. 5)."
  To avoid confusion with actual distribution functions, e.g., the Kappa, instead of using the phase "as illustrated by the distributions of the above and below threshold events", I would suggest the use of phrase similar to: "as illustrated box-plots of Fig.5, constructed by the values of the above and below threshold events".
  **Reply:** *We rephrased the sentence to 'as illustrated by the boxplots depicting the distribution of the above and below threshold events.'*
  **Modification: p.13, l.229**

- **L223-225.** Please remind the reader the selected threshold values.
  **Figure 6 (legend).** I assume that you wanted to write: "… (b) for the **three** catchments in the…."
  **Reply:** *We actually meant to write for the four catchments in the three example regions. The 'three' was added for clarification.*
  **Modification: p.14, caption of Figure 6**

- **L230-233.** The Authors write: "This visual impression of a good performance with respect to the reproduction of spatial correlations in daily discharge data is confirmed by comparing observed and stochastically simulated cross-correlation functions for the catchments in the Pacific Northwest (Fig. 7). Both the shape and magnitude of the cross-correlation functions are well simulated".
  I am afraid that this sentence is needs some refinement, since as with the case of auto-correlation coefficients, Fig. 7 shows a significant deviation of the simulated cross-correlation coefficients from the observed ones. This should be reflected in the manuscript. For the readers convenience, I would also suggest the inclusion of a line depicting the median of all simulations (orange lines). Providing lines for a low and high quantile (say 0.05 and 0.95) would be nice also.

**Reply:** *Figure 7 shows that the ccfs derived from the simulations are very close to those derived from the observations. For a reference we provide Figure 1 in response to reviewer 1, which shows that a model where phases are randomized for each catchment individually would result in a poor reproduction of observed ccfs.*

- **Figure 7.** There is something wrong with the labeling of the panels (i.e., multiple panels are labeled as ii, iii, iv, while some others are completely unlabeled).
  **Reply:** *Each row and column receives a label for the catchment it refers to. These are pairwise plots where cross-correlations are computed for pairs of stations. This specification was added to the caption.*
  **Modification: p.15, caption of Figure 7**

- **239.** The Authors write: "…but also for extreme values as illustrated by the peak-over-threshold (POT) values for the different stations in the three illustration regions (Fig. 9). These results show that besides regional flood co-occurrences, the temporal clustering behavior of events is also reproduced."
  In my view, Figure 9 is not very informative (the y-axis has been omitted intentionally?). Can you please provide an alternative figure, as well as a quantitative metric, quantifying the reproduction of temporal clustering behavior of events?
  **Reply:** *We labeled the y-axis with 'event occurrences'. We think that the figure gives a good impression on whether the temporal clustering features of the observations are reproduced by the simulations. Finding intuitive quantitative measures for this is challenging. However, we computed differences in the mean inter-event duration (i.e. time elapsing between two successive events) of observed and simulated series and the standard deviation of inter-event duration for events where 1, 2, and 3 stations were jointly affected, respectively. We find that over all three regions, relative differences in mean and standard deviation of inter-event duration lies mostly below 10%. However, we find that a visual comparison is most effective here to demonstrate the value of the simulation approach.*
  **Modification: p.15, Figure 9**

- **Discussion section.** In my view all the above points should be discussed in this section, highlighting also the limitations of the presented method. Further to these, it should be noted that the proposed model has been tested for multivariate problems involving 4 processes, as well as the method is capable of generating synthetic timeseries with length equal to the observed one (I haven't read how to handle the case where one wants to generate longer timeseries – is it possible to generate synthetic timeseries with length different than the observed one?).
  **Reply:** *In the discussion we address the limitations of our approach by stating that the approach only allows for the simulation of time series of the same length as the observed series (l 257-259). We would like to highlight here that the modeled range of dependence is also limited to the one in the observed series if one very long time series is generated. We also discuss the disadvantages related to working with distributions fitted to daily values.*

- **L258-259.** The Authors write: "This difference between methods may be related to the fact that the wavelet transform compared to the Fourier transform does not necessitate a transformation to the normal domain, and a back transformation to the domain of the skewed distribution, which has been shown to weaken spatial correlations".
  I think that the answer is is much simpler than the one stated above. The former method is simply designed for the simulation of univariate processes, i.e., not to account for the cross-correlations (or cross-spectrum) among processes. I suggest the Authors to consider more carefully the "mechanics" of the aforementioned methods, and revise the sentence accordingly. Also, as mentioned before, the comment on "weaken spatial correlations"

applies for all "types" of correlations (that emerge from the mapping/transformation from the Gaussian to the actual domain) – not only spatial. Particularly, in the case of stochastic processes, it also applies for the auto-correlation structure of a stationary processes, as well as for the season-to-season correlations of a cyclostationary process (Tsoukalas et al., 2018a, 2017). It also holds for multivariate cases.

**Reply:** *We here specifically focused on the spatial aspect because the focus of our study was on generating spatially consistent time series and event sets. The Fourier-based method results in much weaker spatial correlations than the wavelet-based approach even if phases are randomized in the same way across catchments.*

- **L270-271.** The Authors write: "Thanks to a spatio-temporal model based on phase randomization, temporal short- and long range dependencies, non-stationarities, and spatial dependencies are reproduced."
  Please consider my comments on the parameterization (i.e., number of parameters), as well as on the performance of the model and revise this sentence accordingly.
  **Reply:** *We think that this conclusion is supported by the results presented in our model evaluations throughout the results section, which is also acknowledged by Reviewer 2.*

- **A general comment.** A final comment regards the title of the manuscript, which is: "Stochastic simulation of streamflow and spatial extremes: a continuous, wavelet-based approach". By reading the paper I see that the Authors pay special focus on the reproduction of extremes, but it is not clear to me why this model is different from any other in that aspect (e.g., see those mentioned above)? What makes this model suitable when aiming to reproduce extremes? Other similarly parameterized models behave differently? If yes, why? I believe that a theoretical justification or even an empirical comparison with alternative model(s) would be particularly useful and an added value for the paper.
  Regards,
  Ioannis Tsoukalas
  **Reply:** *We show that PRSim.wave is capable of reproducing spatial dependencies in extremes which has been shown to be challenging in previous studies [Sharma et al., 1997; Caraway et al., 2014]. Our wavelet-based approach has the advantage of being non-parametric in terms of the spatio-temporal model. It avoids assumptions on the distribution and dependence structure of the data. We combine this non-parametric spatio-temporal model with a flexible four-parameter distribution allowing for the representation of a wide range of tail behaviors. We agree that a comparison of different spatial stochastic generators with respect to how they represent spatial extremes would be interesting and valuable. In order to make such a model comparison study beneficial for the community, ideally a broad range of models ranging from continuous to event-based models should be compared as there exists no commonly accepted benchmark/reference model. Such a comparison goes beyond the scope of this manuscript and should be addressed in a separate study.*

**References used in this response to the commentator**

Brunner, M. I., A. Bárdossy, and R. Furrer (2019), Technical note : Stochastic simulation of streamflow time series using phase randomization, *Hydrol. Earth Syst. Sci.*, *23*, 3175–3187, doi:10.5194/hess-23-3175-2019.

Caraway, N. M., J. L. McCreight, and B. Rajagopalan (2014), Multisite stochastic weather generation using cluster analysis and k-nearest neighbor time series resampling, *J. Hydrol.*, *508*, 197–213, doi:10.1016/j.jhydrol.2013.10.054.

Sharma, A., D. G. Tarboton, and U. Lall (1997), Streamflow simulation: a nonparametric approach, *Water Resour. Res.*, *33*(2), 291–308.

---

## Author Response (AR2)

*Dear Prof. Saco,*

*Thank you for acknowledging the positive development of our manuscript. We added the few methodological specifications requested by the reviewer and extended the discussion on limitations and future directions. We hope that you find the updated version suitable for publication in HESS. Thank you very much for your efforts with our manuscript.*

*Kind regards,*

*Manuela Brunner*

**Reviewer 1**

GENERAL COMMENTS
Authors have addressed most comments in the previous review. I have some follow up comments and if these issues are addressed, I would suggest acceptance.

**Reply:** *Thank you for your comments, which we address point by point below.*

Comments:
COMMENT1: In Figure 4, the simulated discharges show overestimated ACF in most realizations. Any particular reason for this? Similar for the Figure 7, the simulated ccf captures the pattern but seems to overestimate among investigated stations.

**Reply:** *For the particular station shown in Figure 4i, we indeed observe an overestimation of the ACF at time lags > 10 days. However, we do not observe such an overestimation systematically across stations. For other stations, ACFs may not be perfectly reproduced at other time lags. It is hard to tell why this station shows that particular deviation from the observed ACF.*

COMMENT2: There is little discussion on the results of Figure 8. It is supposed to be discussed at Line 252 with comparison with Fig. A2. First, the stochastic simulations, are they 100 realizations or just one ensemble mean/median simulation? No matter which case, simulated variogram is underestimated while in Fig. A2 the phase randomized for each catchment, the simulated variograms are mostly overestimated. I would suggest authors include a discussion on this issue.

**Reply:** *Thank you for pointing out the lack of mentioning Figure 8. We indeed mixed up the Figure references. We added a reference to Figure 8 and specify that 'The good performance in terms of reproducing the general spatial dependence structure in the data can be generalized to other regions as shown by a comparison of observed and simulated variograms indicating a slight overestimation of spatial dependence for very long distances (> 30 degrees, ca. 3000 km; Fig. 8). We then go on to discuss the underestimation of spatial dependence in the case phases are not jointly randomized across stations: 'In the case of individual randomization, spatial dependence is not event captured for very short distances of a few kilometers.' We clarify in the caption of Figure 8 that the 100 variograms are derived from 100 model runs: 'Comparison of observed variogram (black) with 100 variograms derived from the 100 simulation runs (orange).'*
**Modification: p.14 l.253-254 and 256-257; Figure 8 caption**

COMMENT3: If there is a generally good agreement between observed and simulated spatial dependence in Figure 11 using F-madogram, then the simulated variogram with phase randomized for each catchment in Fig.A2 can be seen having a good agreement with observations as well.

Another doubt is that the observed and simulated spatial dependence seems like have different range in the x-axis. Why?

**Reply:** *Thank you for indicating the need for clarification. Figure 11 shows the F-madogram, which is a measure for the strength of spatial dependence in extremes. It takes values between 0 and 1, which was now specified in the text. In contrast, Figure A2 shows variograms, which are a measure of overall dependence not focused solely on extremes. The variogram takes values between 0 and inf. We mention in the text that the variogram is used as a measure for the general dependence structure (l. 211-214) and the F-madogram as a measure for the spatial dependence in extremes (l.219-212). Figure A2 can therefore not be compared to Figure 11 and should be rather compared to Figure 8, which shows the variograms derived with the model where phases are jointly randomized across all stations. Figures 8 and A2 share the same x-axes and are directly comparable. However, we note that not randomizing phases jointly across catchments, can not capture observed variograms.*
**Modification: p.9, l.221-222**

COOMENT4: In this work, authors have shown that the proposed technique to generate multisite hydrological extremes is able to capture both spatial dependencies and non-stationarities. The logic is very similar to the work of Jiang et al. (2020), where they modify the variance structure of the variables in the frequency domain to match the spectral attributes of one another. A discussion of the similarities in the two should be included. Although the focus here is to randomize the phases in the frequency domain, it is worth checking the difference of wavelet variance (Percival, 1995) before and after randomization, and this might be the reason that the overestimation and/or underestimation in your temporal and spatial dependence analysis. It is not necessary to include further results in the current manuscript but a short discuss of limitation and future development might be helpful.

**Reply:** *Thank you for pointing out these two references. We agree that instead of the phases, the wavelet coefficients could be randomized. As mentioned in the introduction, there have been previous attempts to do so (l.81-89). However, it has been shown that a randomization of wavelet coefficients may not preserve non-stationarities and long-term periodicities as some block-bootstrap approach is usually used for resampling [Breakspear et al., 2003]. We did check the reproduction of the normalized average power (Figures 4 i and ii d) which averages the wavelet amplitudes over time and shows which temporal scales explain the biggest part of the variance. Our results show that this average power is generally well reproduced by the stochastic model. We did not include Jiang et al. (2020) as a reference because they deal with a prediction problem rather than stochastic simulation and use a discrete instead of a continuous wavelet transform and we therefore think that establishing a link between the two approaches is a bit farfetched. The manuscript includes a Discussion section which discusses the limitations of and future directions for the modeling approach. We extended this section by stating: 'However, there is a slight tendency of the model to overestimate spatial dependence in extremes.' and provide directions on how the representation of spatial dependencies could be further improved: 'A further improvement of the representation of cross-correlations and spatial dependencies may be achieved by using phase annealing, which modifies the phases in an iterative way in order to optimize certain statistics but increases the computational effort Hörning and Bárdossy [2018].' We also extended the discussion on potential future work by stating: 'Potential future extensions also include the consideration of covariates to more explicitly model non-stationarities.'*
**Modification: p.19: l.278-279, l. 283-285, l.295**

JIANG, Z., SHARMA, A. & JOHNSON, F. 2020. Refining Predictor Spectral Representation Using

Wavelet Theory for Improved Natural System Modeling. Water Resources Research, 56, e2019WR026962.

PERCIVAL, D. P. 1995. On estimation of the wavelet variance. Biometrika, 82, 619-631.

**References used in this response to the reviewer**

[revised manuscript text omitted]